# Strategy-dependent effects of working-memory limitations on human perceptual decision-making

Kyra Schapiro[1]*, Krešimir Josić[2,3], Zachary P Kilpatrick[4,5], Joshua I Gold[1]

[1]Department of Neuroscience, University of Pennsylvania, Philadelphia, United States; [2]Department of Mathematics, University of Houston, Houston, United States; [3]Department of Biology and Biochemistry, University of Houston, Houston, United States; [4]Department of Applied Mathematics, University of Colorado Boulder, Boulder, United States; [5]Institute of Cognitive Science, University of Colorado Boulder, Boulder, United States

**Abstract** Deliberative decisions based on an accumulation of evidence over time depend on working memory, and working memory has limitations, but how these limitations affect deliberative decision-making is not understood. We used human psychophysics to assess the impact of working-memory limitations on the fidelity of a continuous decision variable. Participants decided the average location of multiple visual targets. This computed, continuous decision variable degraded with time and capacity in a manner that depended critically on the strategy used to form the decision variable. This dependence reflected whether the decision variable was computed either: (1) immediately upon observing the evidence, and thus stored as a single value in memory; or (2) at the time of the report, and thus stored as multiple values in memory. These results provide important constraints on how the brain computes and maintains temporally dynamic decision variables.

*For correspondence:
kaschapiro@aol.com

## Editor's evaluation

This paper employs sophisticated modeling of human behavior in well-controlled tasks to study how limitations of working memory constrain decision-making. Because both are key cognitive processes, that have so far largely been studied in isolation, the paper will be of broad interest to neuroscientists and psychologists. The observed working memory limitations support previous findings and extend them in critical ways.

## Introduction

Many perceptual, memory-based, and reward-based decisions depend on an accumulation of evidence over time (*Brody and Hanks, 2016*; *Gold and Shadlen, 2007*; *Ratcliff et al., 2016*; *Shadlen and Shohamy, 2016*; *Summerfield and Tsetsos, 2012*). This dynamic process, which can operate on timescales ranging from tens to hundreds of milliseconds for many perceptual decisions to seconds or longer for reward-based and other decisions (*Bernacchia et al., 2011*; *Gold and Stocker, 2017*), requires working memory to maintain representations of new, incoming evidence and/or the aggregated, updating decision variable. Working memory is constrained by capacity and temporal limitations (*Bastos et al., 2018*; *Cowan et al., 2008*; *Funahashi et al., 1989*; *Oberauer et al., 2016*; *Panichello et al., 2019*; *Ploner et al., 1998*; *Schneegans and Bays, 2018*; *White et al., 1994*) that, in principle, could also constrain decision performance. Several previous studies failed to identify such constraints on decisions that depend on working memory, but those studies used tasks involving

 

**eLife digest** Working memory, the brain's ability to temporarily store and recall information, is a critical part of decision making – but it has its limits. The brain can only store so much information, for so long. Since decisions are not often acted on immediately, information held in working memory 'degrades' over time. However, it is unknown whether or not this degradation of information over time affects the accuracy of later decisions.

The tactics that people use, knowingly or otherwise, to store information in working memory also remain unclear. Do people store pieces of information such as numbers, objects and particular details? Or do they tend to compute that information, make some preliminary judgement and recall their verdict later? Does the strategy chosen impact people's decision-making?

To investigate, Schapiro et al. devised a series of experiments to test whether the limitations of working memory, and how people store information, affect the accuracy of decisions they make. First, participants were shown an array of colored discs on a screen. Then, either immediately after seeing the disks or a few seconds later, the participants were asked to recall the position of one of the disks they had seen, or the average position of all the disks. This measured how much information degraded for a decision based on multiple items, and how much for a decision based on a single item. From this, the method of information storage used to make a decision could be inferred.

Schapiro et al. found that the accuracy of people's responses worsened over time, whether they remembered the position of each individual disk, or computed their average location before responding. The greater the delay between seeing the disks and reporting their location, the less accurate people's responses tended to be. Similarly, the more disks a participant saw, the less accurate their response became. This suggests that however people store information, if working memory reaches capacity, decision-making suffers and that, over time, stored information decays.

Schapiro et al. also noticed that participants remembered location information in different ways depending on the task and how many disks they were shown at once. This suggests people adopt different strategies to retain information momentarily.

In summary, these findings help to explain how people process and store information to make decisions and how the limitations of working memory impact their decision-making ability. A better understanding of how people use working memory to make decisions may also shed light on situations or brain conditions where decision-making is impaired.

binary choices that may be relatively insensitive to known working-memory limitations (*Liu et al., 2015*; *Waskom and Kiani, 2018*). It remains unclear if and how working-memory limitations affect decisions that require interpreting and storing continuously valued quantities whose representations are known to degrade over time (*Ploner et al., 1998*; *Schneegans and Bays, 2018*; *Wei et al., 2012*; *White et al., 1994*).

To better understand how working-memory limitations affect decision-making, we examined how humans made decisions that required interpreting and storing continuously valued visuo-spatial information (visual target locations) that is sensitive to capacity and temporal limitations of working memory (*Bastos et al., 2018*; *Funahashi et al., 1989*; *Panichello et al., 2019*; *Ploner et al., 1998*; *Schneegans and Bays, 2018*; *White et al., 1994*). Specifically, we required participants to indicate a remembered spatial location that was informed by one or more briefly presented visual stimuli ('disks'; *Figure 1*) after a variable delay. We compared the effects of variable set size and delay when the remembered location corresponded to either: (1) the perceived location (angle) of a specific disk, identified at the time of interrogation, which is a design that has been used previously (*Ploner et al., 1998*; *Schneegans and Bays, 2018*; *Wei et al., 2012*; *White et al., 1994*); or (2) the computed mean angle of a set of multiple disks, which is a form of continuous decision variable whose sensitivity to working-memory limitations has not been examined in detail. Additionally, we examined the effects of working-memory limitations on computed locations under two conditions that are representative of certain decision-making tasks. The first was a 'simultaneous' condition in which all disks (and thus all information) were presented at once. The second was a 'sequential' condition in which one disk was presented later than the others. This condition required participants to adjust to a within-trial change of available decision-relevant information, typifying decisions that require evidence accumulation over time.

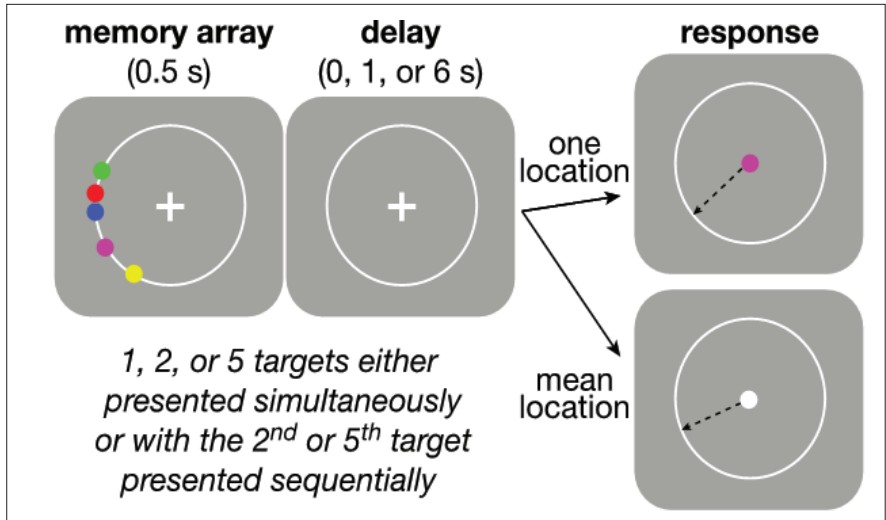

**Figure 1.** Behavioral task. Participants were asked to maintain visual fixation on the center cross while an array of colored disks was presented for 0.5 s, followed by a variable delay and finally the presentation of a visual cue whose color was either: (1) the same as one of the disks, indicating that the participant should use the mouse to mark the remembered location of that disk ('Perceptual' trial) or (2) white, indicating that the participant should mark the mean angle of the array ('Computed' trial). Perceptual and Computed trials were presented in separate, signaled blocks. On Perceptual trials, participants did not know in advance which disk would be probed on any given trial. The number of disks and length of the delay period were varied randomly within each block. Blocks were also defined by the temporal presentation of the disks. In 'Simultaneous' blocks, all disks were presented at once, whereas in 'Sequential' blocks, the final disk (always the most counter-clockwise of all disks presented on that trial) was presented midway through the variable delay. In all blocks, the disks always had the same clockwise ordering by color, as depicted in the 'memory array' graphic above, to minimize binding errors between color and location in the Perceived blocks.

For spatial working-memory tasks, the precision of working memory for perceived spatial locations is often well described by diffusion dynamics (*Compte et al., 2000*; *Kilpatrick, 2018*; *Kilpatrick et al., 2013*; *Laing and Chow, 2001*) that are commonly implemented in 'bump-attractor' models of working memory (*Compte et al., 2000*; *Constantinidis et al., 2018*; *Laing and Chow, 2001*; *Riley and Constantinidis, 2016*; *Wei et al., 2012*; *Wimmer et al., 2014*). Our analyses built on this framework by examining memory diffusion dynamics for the different task conditions and potential decision strategies. For the conditions we tested, most participants' behavior was well fit by one of two distinct strategies, each with its own constraints on decision performance based on different working-memory demands. The first strategy was to compute the decision variable (mean disk angle) immediately upon observing the evidence (individual disk angles), and then store that value in working memory in a manner that, like for the memory of a single perceived angle, could be modeled as a single particle with a particular diffusion constant (Average-then-Diffuse model; AtD). The second strategy was to maintain representations of all disk locations in working memory, modeled as separate diffusing particles, and then to combine them into a decision variable only at the time of the decision (Diffuse-then-Average model; DtA). Such a strategy results in an effective diffusion constant for the average that is inversely related to the number of items. Our results show that like perceived locations, memory for computed mean locations degraded with increased set size (of relevant information), and delay between presentation and report. However, the degree of degradation depended on the strategy used to compute the decision variables, implying that multiple, strategy- and task-dependent effects of working-memory should be considered in the construction of future neural and computational models of decision-making.

## Results

We measured the ability of human participants to remember spatial angles as a function of set size (1, 2, or 5 disks), delay duration (0, 1, or 6 s), and task context (Perceived or Computed blocks). Specifically,

we measured the error between reported and probed angles as a proxy for working-memory representations and inferred rates of memory degradation (diffusion constants) from the increase in variance of these errors over time within a framework of diffusing-particle models. Below we first describe the model framework, detailing its key assumptions and predictions. We next describe results from Simultaneous conditions, in which all items were presented simultaneously at the beginning of each trial, which demonstrate how capacity and temporal constraints on working memory relate to the accuracy of computed decision variables. We then describe results from Sequential conditions, in which one item was presented after the others in each trial, which demonstrate how capacity and temporal constraints affect the process of evidence integration over time.

## Diffusing-particle framework and predictions

Within our diffusing-particle framework, the memory of an item is represented by the location of a diffusing particle. This representation allows us to quantify the corruption (i.e., reduced precision) of the memory by two distinct sources of noise. The first is described by a static, additive term ($\eta_1$) that encompasses all potential one-time noise sources within a trial including noise associated with the sensory encoding and the motor response. The second is the dynamic degradation of memory precision over time that is modeled as the diffusion of the particle (*Figure 2a*). This diffusion corresponds to an increase in variability over time that is linear, with a slope equal to the diffusion constant ($\sigma_1^2$; *Figure 2b*). Consistent with past modeling studies (*Bays et al., 2009*; *Brady and Alvarez, 2015*; *Koyluoglu et al., 2017*; *Wei et al., 2012*), we accounted for the decrease in working-memory fidelity with item load by incorporating item number (*N*) dependence into both the static noise term ($\eta_N$) and the diffusion constant of each particle ($\sigma_N^2$; *Figure 2b*).

We extended this framework to account for working-memory representations of values computed from multiple stimuli, namely their average location, via two primary models (these models also served as the basis for more complex extensions, including mixtures of the two models, used to account for the Sequential condition detailed in subsequent sections). In the first, called the AtD model, the average is calculated immediately upon observing the evidence and then stored as a single particle in working memory. This model has its own static noise term that includes variability in estimates of the mean of *N* items ($\eta_{MN}$) and then assumes that the single estimate held in working memory diffuses with the same diffusion constant as a single perceived item ($\sigma_{MN}^2=\sigma_1^2$; see the parallel purple and black lines in *Figure 2b* and the overlapping lines in *Figure 2c*). In the second, called the DtA model, the memories of all constituent items are maintained and then combined into a decision variable (the average) only at the time of the response. This model assumes an effective diffusion constant for the reported average that is related to $\sigma_N^2$ by the inverse of the number of items ($\sigma_{MN}^2=\sigma_N^2/N$) because averaging over *N* random variables with a variability of $\sigma_N^2$ results in a random variable with variability $\sigma_N^2/N$.

The ability to distinguish these two models depends on their relative ability to capture specific changes in error over time in the report of the average, which in turn depends on the relationship between the diffusion constant of a single item and multiple items. We describe this relationship as $\sigma_N^2=\sigma_1^2*N^A$, where *A* is a constant for a given participant and set size that describes the cost to store *N* items in memory. Because of the previously described relationships between $\sigma_1^2$, $\sigma_N^2$, and $\sigma_{MN}^2$, it is therefore also true that (the following constraints were enforced when determining best-fitting values of *A* using data from both Perceived and Computed blocks): (1) in the AtD model, $\sigma_N^2=\sigma_{MN}^2*N^A$ (i.e., the diffusion constant describing memory degradation over time for *N* Perceived items held in working memory is proportional to the diffusion constant describing memory degradation over time for the one Computed value held in working memory); and (2) in the DtA model, $\sigma_{MN}^2=\sigma_1^2*N^A/N$ (i.e., the diffusion constant describing memory degradation over time for *N* Computed items held in working memory is proportional to the diffusion constant describing memory degradation over time for one Perceived item held in working memory). For a given static noise level and $\sigma_1^2$, the *A* parameter dictates whether AtD or DtA has a lower $\sigma_{MN}^2$ and thus results in lower memory loss over time (*Figure 2c*). Specifically, when *A*<1, DtA has a lower $\sigma_{MN}^2$ and less variable responses because the averaging over multiple diffusing items counteracts the greater total noise of having many items. When *A*=1, the additional noise cost of each individual point in DtA exactly balances the effect of averaging, such that AtD and DtA have equal $\sigma_{MN}^2$ and equal levels of accuracy and thus are indistinguishable (*Figure 2—figure supplement 1* shows the models becoming increasingly indistinguishable as A approaches 1). When *A*>1, the additional cost of storing multiple items outweighs the effect

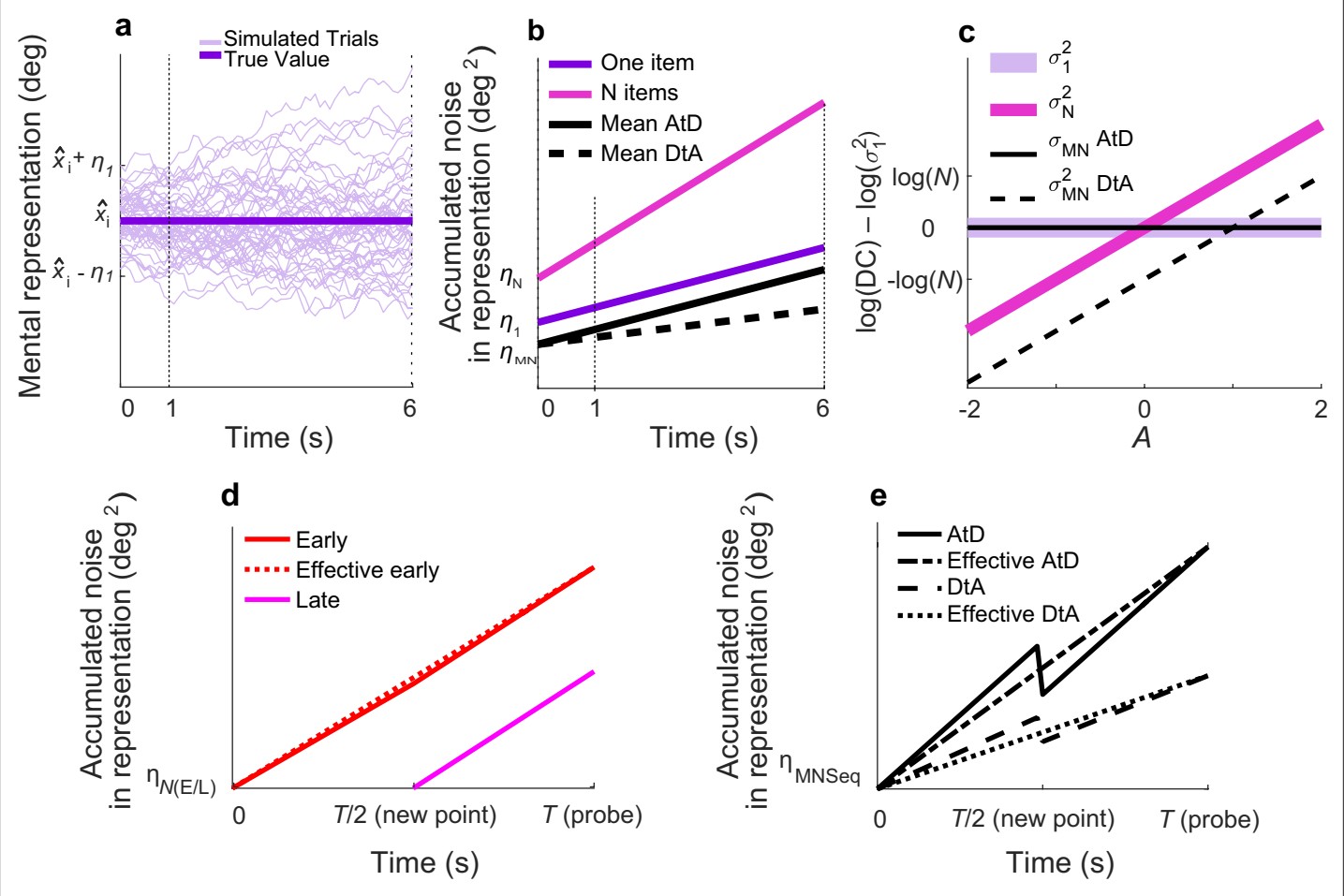

**Figure 2.** Diffusion model and predictions for different strategies. (**a**) Fifty simulated trials of the representation of a single memorandum, $\hat{x}_i$, corrupted by a static noise term representing sensory and motor noise ($\eta_1$) and time-dependent noise (increasing variance corresponding to decreasing memory precision) modeled as Brownian diffusion. At time $t$, the report for one item, $r_{t,1}$, is the location of the particle. (**b**) Linear accumulation of noise (variance) for single or multiple Perceived items (colors, as indicated) or Computed mean values using two different strategies (solid vs. dashed black lines, as indicated). Memory representations of $N$=1, 2, or 5 items have initial, additive error $\eta_N$ and diffuse over time with diffusion constant $\sigma_N^2$; thus, variance at time $t=\eta_N+t*\sigma_N^2$. For the Average-then-Diffuse (AtD) model, the average is calculated immediately and stored as a single value. Thus, the diffusion constant of a Computed mean of $N$ items is the same as for one item ($\sigma_{MN}^2=\sigma_1^2$; parallel purple and black lines), although $\eta_1$ and $\eta_{MN}$ may not be equal. For the Diffuse-then-Average (DtA) model, all items are stored until the probe time. Thus, the effective $\sigma_{MN}^2$ is $1/N$th of $\sigma_N^2$. (**c**) Relationship between $A$ and log differences of diffusion constants for various set sizes and models. $\sigma_1^2$ is independent from $A$ and equal to $\sigma_{MN}^2$ under AtD. $\sigma_N^2$ is linear with $A$ in log space with respect to ($\sigma_1^2$) because $\log(\sigma_N^2)-\log(\sigma_1^2)=A*\log(N)$. $\sigma_{MN}^2$ is linear with $A$. DC=Diffusion Constant. (**d**) Accumulation of noise for Perceived items presented sequentially. When the new (Late) point is added at time $T/2$, the diffusion constant for previously presented items (Early) changes slightly because of the increased load. Early and Late items for set size $N$ have encoding noise $\eta_{NE}$ and $\eta_{NL}$, respectively, represented by $\eta_{(E/L)}$. The 'effective Early' trace shows the net gain in variance over time that would be expected when sampling the error only at a single time $T$, as we did. (**e**) Accumulation of noise for Computed items in the Sequential condition for both models. The encoding noise for the mean of $N$ items is represented by $\eta_{MNSeq}$. At time=$T/2$, the final point is averaged, causing a change in the diffusion constant. The 'effective' lines represent the measured change in variance over time one would measure when recording only at $T$. Here $N$=5, $A$=0.5.

The online version of this article includes the following figure supplement(s) for figure 2:

**Figure supplement 1.** Identifiability of AtD and DtA models as a function of the $A$ parameters.

of averaging, and AtD produces a lower $\sigma_{MN}^2$ and less variable responses than DtA. A summary of all framework variables can be found in **Table 1**.

To summarize, our two models describe two different possible ways for decision-relevant information to be stored in working memory prior to executing a decision. The different storage strategies result in different patterns of memory degradation, corresponding to trial-to-trial variability (imprecision) of decision reports that increase as a function of the length of the within-trial delay period. For

**Table 1.** Descriptions of all framework and model parameters.

Fit parameters are shown on the top. Derived parameters used in other analyses and descriptions are shown on the bottom. Variations used to model Sequential conditions are shown to the right.

| Fit model parameters (units) | Description | Variants for sequential conditions | Description |
|---|---|---|---|
| $\eta_1$ (deg$^2$) | Static noise for a single item | | |
| | | $\eta_{NE}$ (deg$^2$) | Static noise for Early items |
| $\eta_N$ (deg$^2$) | Static noise for $N$ items | $\eta_{NL}$ (deg$^2$) | Static noise for Late items |
| $\eta_{MN}$ (deg$^2$) | Static noise for mean of $N$ items | $\eta_{MN\_seq}$ (deg$^2$) | Static noise for mean of $N$ items in Sequential Conditions |
| $\sigma_1^2$ (deg$^2$/s) | Diffusion constant tor single item | | |
| $A$ (no units) | Diffusion cost of storing $N$ items | | |
| **Derived descriptive terms** | | | |
| | | $\sigma_{NE}^2$ (deg$^2$/s) | Diffusion constant for Early items |
| $\sigma_N^2$ (deg$^2$/s) | Diffusion constant for $N$ items$=\sigma_1^2*N^A$ | $\sigma_{NL}^2$ (deg$^2$/s) | Diffusion constant for Late items |
| | Diffusion constant for mean of $N$ items | $\sigma_{MN\_Seq}^2$ (deg$^2$/s) | Diffusion constant for mean of $N$ items in Sequential Conditions |
| $\sigma_{MN}^2$ (deg$^2$/s) | AtD: $=\sigma_1^2$ DtA: $=\sigma_1^2*N^A/N$ $=\sigma_N^2/N$ | | |

the AtD model, the individual pieces of information are immediately combined into a single decision variable that is then stored in memory. Thus, the rate of degradation of an estimated average is identical to the rate of degradation of a single item. In contrast, for the DtA model, all of the relevant pieces of information are stored in memory and then combined only at the time of indicating the decision. Thus, the rate of degradation of an estimated average is inversely proportional to the rate of degradation of each item held in memory. We used fits of these models to performance data from individual participants to distinguish different patterns of memory degradation and therefore different storage strategies.

## Simultaneous condition behavior

When all disks were presented simultaneously, performance was consistent with several key predictions of the particle model. Specifically, the difference in reports of Perceived spatial angles and the true probed location (i.e., the response error) tended to be unbiased, in that the mean error across participants was not reliably different from 0 (*Figure 3a*, full distributions in *Figure 3—figure supplement 1*, individual participant mean errors in *Figure 3—figure supplement 2*). However, the variance of these errors increased roughly linearly over time (*Figure 3c*), like the location of a diffusing particle or bump attractor (*Compte et al., 2000*; *Kilpatrick, 2018*; *Kilpatrick et al., 2013*; *Laing and Chow, 2001*). This error variance depended systematically on set size (*Figure 3c*). However, the change in error variance over time (slope of variance increase) did not depend on set size (ANOVA, significant effect of set size, $F_{(2,32)}=83.87$, $p=1.88\mathrm{e}{-13}$, and delay, $F_{(2,32)}=29.55$, $p=5.37\mathrm{e}{-08}$, but no significant interaction between set size and delay, $F_{(4,64)}=1.36$, $p=0.256$). Errors in reports of Computed (i.e., inferred mean) spatial angles relative to true mean angles showed similar trends, albeit with a much weaker dependence on the number of items. Specifically, Computed angle reports were also unbiased (mean error from the true value was not reliably different from 0; *Figure 3b*, *Figure 3—figure supplements 1 and 3*) but degraded (became more variable) with a roughly linear increase in variance over time (*Figure 3d*). Error variance in the report of the Computed average was higher at higher set sizes (set size 5 had higher variances), but the rate of degradation in accuracy did not depend on set size (ANOVA, significant effect of set size, $F_{(2,32)}=13.53$, $p=5.515\mathrm{e}{-5}$, and delay, $F_{(2,32)}=130.79$, $p=4.441\mathrm{e}{-16}$, but not their interaction, $F_{(4,64)}=0.538$, p=0.708).

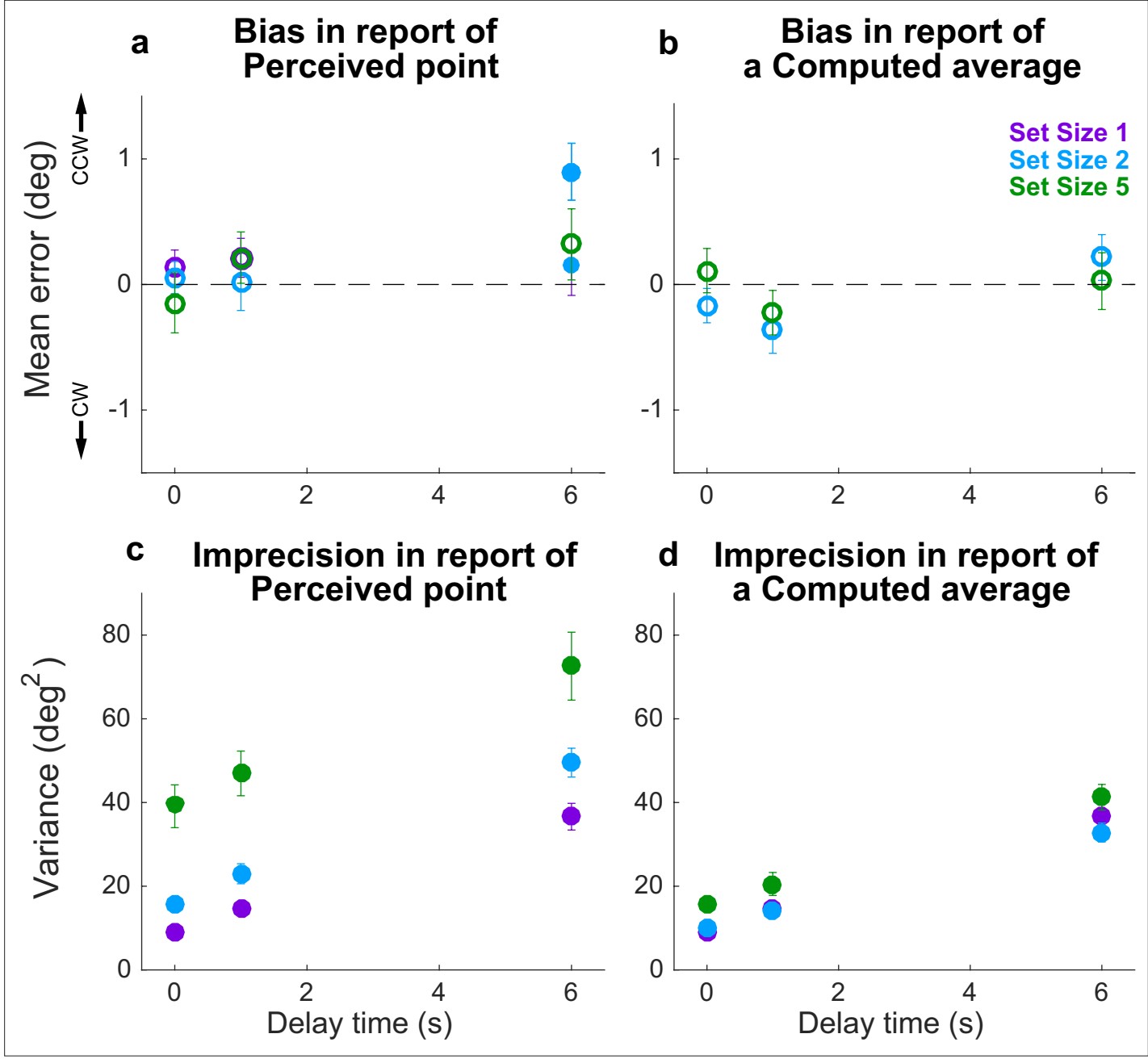

**Figure 3.** Behavioral summary for the Simultaneous condition. (**a**) Mean Perceptual error for different set sizes (colors, as indicated) and delay times (abscissa). Filled points indicate two-tailed *t*-test for $H_0$: mean=0, $p<0.05$. (**b**) Mean Computed (inferred mean) error for different set sizes (colors, as indicated) and delay times (abscissa). For all tests, mean error was not significantly different from 0 ($p>0.05$; open circles). (**c**) Variance in Perceptual errors, plotted as in (**a**). (**d**) Variance in Computed (mean) errors, plotted as in (**b**). In each panel, points and error bars are mean ± SEM across participants.

The online version of this article includes the following figure supplement(s) for figure 3:

**Figure supplement 1.** Full error distributions in Simultaneous conditions.

**Figure supplement 2.** Participant-wise mean response error in the Simultaneous Perceived condition.

**Figure supplement 3.** Participant-wise mean error in the Simultaneous Computed condition.

**Table 2.** Summary of model fits for the Simultaneous condition.

Parameters are: (1) $\eta_1$, static noise for a single item; (2) $\eta_N$, static noise for $N$ items; (3) $\eta_{MN}$, static noise for the mean of $N$ items; (4) $\sigma_1^2$, diffusion constant for a single item; and (5) A, diffusion cost for additional items. For each parameter, the maximum-likelihood estimates are given for the participants' best fit by the indicated model (mean ± SEM across participants). * indicates *t*-test for $H_0$: difference between the mean values of each parameter across models for a given set size=0, p<0.05.

| | Set size (N) | Number best-fit participants | $\eta_1$ | $\eta_N$ | $\eta_{MN}$ | $\sigma_1^2$ | A |
|---|---|---|---|---|---|---|---|
| AtD | 2 | 8 | 10.79±1.45 | 16.49±2.67 | 9.39±1.07 | 4.85±0.44 | 0.0892±0.24 |
| | 5 | 14 | 9.80±1.18 | 36.88±5.33 | 14.79±1.50 | 4.34±0.47 | 0.0051±0.07* |
| DtA | 2 | 9 | 8.22±1.35 | 14.16±3.13 | 10.45±2.09 | 3.67±0.58 | 0.61±0.22 |
| | 5 | 3 | 7.63±0.30 | 45.49±17.03 | 21.10±7.59 | 5.14±0.28 | 0.49±0.14* |

## Simultaneous condition model fits

To better understand the effects of delay and set size on working-memory representations of Perceived and Computed angles for individual participants, we fit the AtD and DtA models separately to data from each set size (N=2 or 5) condition and participant (*Table 2*; the two models each had the same number of free parameters and thus were compared using the log-likelihoods obtained from the fits). The fitting procedures for both models used data from all trials from Perceived set size one and set size $N$ conditions, and from Computed set size $N$ conditions. In general, both strategies were used by our participants in each of the two set-size conditions (for set sizes 2, 8, and 9 participants were best fit by the AtD and DtA models, respectively; for set sizes 5, 14, and 3 participants were best fit by the AtD and DtA models, respectively).

Because the two models are indistinguishable when $A=1$ (i.e., $\sigma_{MN}^2 = \sigma_1^2 = \sigma_1^2 * N^A/N = \sigma_N^2/N$), we further analyzed the best-fitting values of $A$. Across our participants, the 95% confidence intervals (CIs) for $A$ (determined from the SEM values shown in *Table 2*) did not overlap with 1, supporting the distinguishability of the two models, on average (although not for each individual participant; *Figure 4—figure supplement 1a,b*). Moreover, best-fitting values of $A$ were similar when they were estimated separately from Perceived versus Computed blocks for DtA participants (two-sided test for $H_0$: difference in mean best-fitting values across participants=0, p=0.895 and 0.452 for set sizes 2 and 5, respectively; note that $A$ is not defined for the AtD model on Computed blocks alone), which supports our modeling assumption that $A$ (the cost of storing $N$ items in memory) is roughly the same in the two blocks. For the participants' best fit by the AtD model, the mean, best-fitting values of $A$ for both set sizes were close to 0. These values were consistent with the lack of interaction between set size and delay in the Perceptual ANOVA in *Figure 3c* (because in this model, $\sigma_N^2 = \sigma_1^2$ when $A=0$). Conversely, for the participants' best fit by the DtA model, the mean, best-fitting values of $A$ were slightly higher (the difference between each parameter between groups for a given set size was only significantly different from 0 for $A$ at set size 5; see *Table 2*). These values were consistent with the lack of interaction between set size and delay in the Computed ANOVA in *Figure 3d* (because in this model, $\sigma_{MN}^2 = \sigma_1^2 * N^A/N$, which becomes less dependent on $N$ when $A$ approaches 1).

## Simultaneous condition model validation

When $A$ differs from 1, AtD and DtA make distinct assumptions about the diffusion constant relationships between either single (AtD) or multiple (DtA) Perceived angles(s) versus a Computed average angle (*Figure 2b and c*). We used these assumptions to validate whether the better-fitting model and best-fit parameters for a given participant at a given set size were likely to produce the participant's behavior. Specifically, the AtD model assumes that the diffusion constant for a single Perceived angle and for a Computed average angle are the same because both involve the memory of a single value (*Equation 9*). In contrast, the DtA model assumes that the diffusion constant for a Computed average angle is $1/N$th the diffusion constant for $N$ items, because all $N$ items are held in memory prior to averaging (*Equation 10*). We analyzed how consistent these assumptions were with the behavioral data

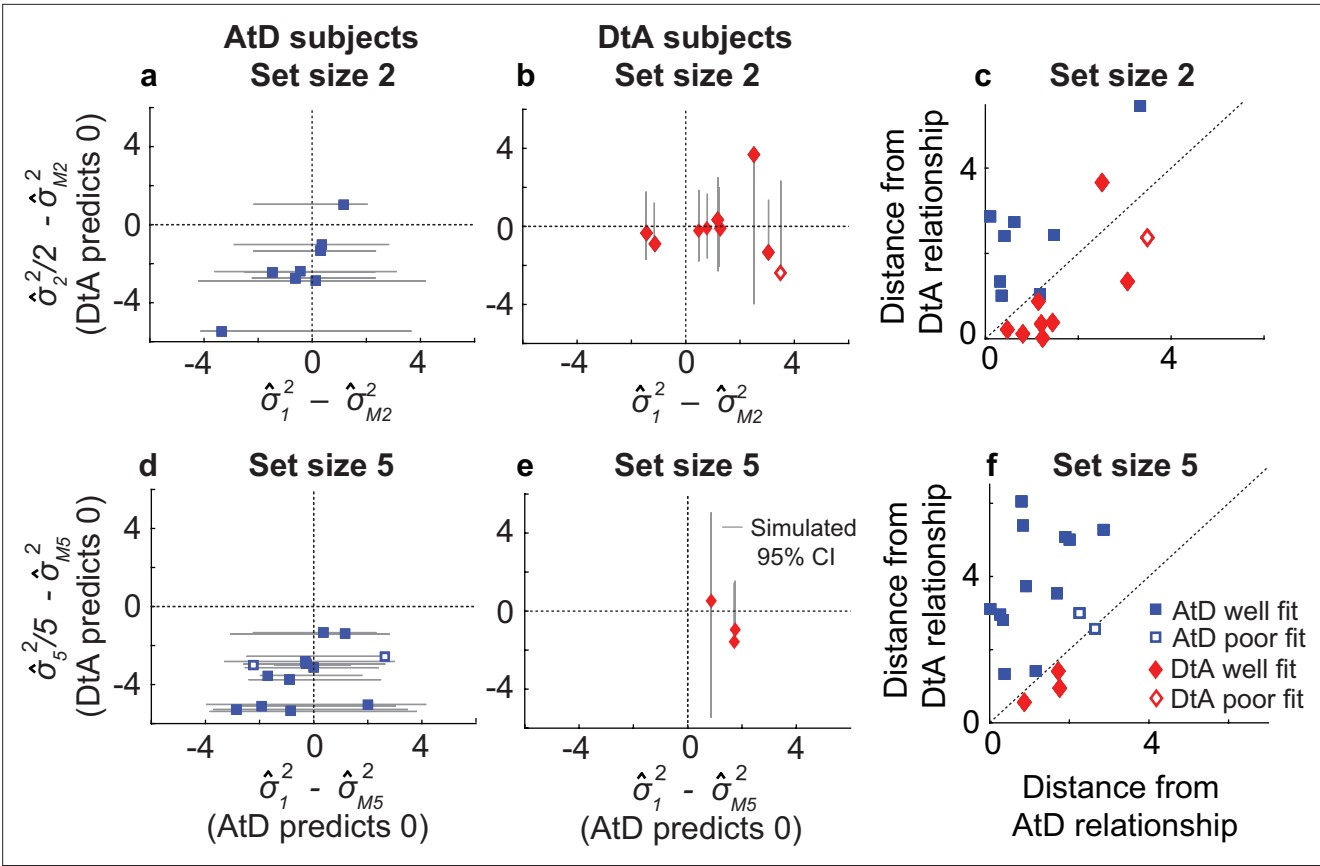

**Figure 4.** Comparisons of empirical and model-based diffusion constant relationships for the Simultaneous condition. In (**a, b, d, e**), the abscissa shows the difference between: (1) empirical estimates of the diffusion constant for a Computed value measured by fitting a line to measured variance as a function of delay time for set size 2 ($\hat{\sigma}_{M2}^2$, **a, b**) or 5 ($\hat{\sigma}_{M5}^2$, **d, e**), and (2) the empirical estimates of the diffusion constant for a single Perceived value ($\hat{\sigma}_1^2$). The AtD model predicts a difference of 0. The ordinate shows the difference between: (1) the empirical estimate of Computed diffusion constants $\hat{\sigma}_{M2}^2$ or $\hat{\sigma}_{M5}^2$, and (2) the empirical estimates of the diffusion constant for multiple Perceived values ($\hat{\sigma}_2^2$ or $\hat{\sigma}_5^2$) divided by the number of items. The DtA model predicts a difference of 0. Each point was obtained using data from individual participants, separated by whether they were best fit by the AtD (**a, b**) or DtA (**d, e**) model for the given set-size condition. Lines represent 95% confidence intervals (CIs) computed by simulating data using the best-fit parameters for the given fit and repeating the empirical estimate comparison procedure. Closed symbols indicate participants who fell within the 95% CI for their best-fit model. (**e, f**) Distance of each participant's empirically estimated diffusion constant relationships from those predicted by AtD or DtA (i.e., distances from the x=0 and y=0 lines, respectively, in (**a, b, d, e**)), for set sizes 2 (**c**) and 5 (**f**).

The online version of this article includes the following figure supplement(s) for figure 4:

**Figure supplement 1.** Participant-specific estimates of *A* from the Simultaneous condition for set sizes 2 (**a**) and 5 (**b**).

(*Figure 4*). For each participant, we fit a line to the measured error variances as a function of delay for a given set size in both Perceived and Computed blocks to estimate the change in variance over time (the empirical diffusion constant estimates: $\hat{\sigma}_1^2$, $\hat{\sigma}_N^2$, $\hat{\sigma}_{MN}^2$, where *N*=2 or 5 for the two set sizes). We then compared the differences of these empirical estimates to the differences predicted between diffusion constants by the best fit model for a given participant.

In general, the participant data conformed to the model predictions of the best-fit model for that participant, despite substantial individual variability. For participants whose data were best fit by the AtD model, empirical estimates of the diffusion constant ($\hat{\sigma}_{MN}^2$) from Computed blocks tended to be similar to the empirical estimates of the diffusion constant for a single Perceptual point ($\hat{\sigma}_1^2$; *Figure 4a and b*). Specifically, for all but two participants, the empirical diffusion constant differences fell within the 95% CI of simulated distribution. Likewise, for participants whose data were best fit by the DtA model, empirical estimates of the diffusion constant ($\hat{\sigma}_{MN}^2$) from Computed blocks tended to be similar to the empirical estimates of the diffusion constant for multiple Perceptual items divided by the set size ($\hat{\sigma}_N^2/N$; *Figure 4d and e*). Specifically, for all but one participant, empirical diffusion constant differences fell within the 95% CI of the simulated distribution. These analyses, which are

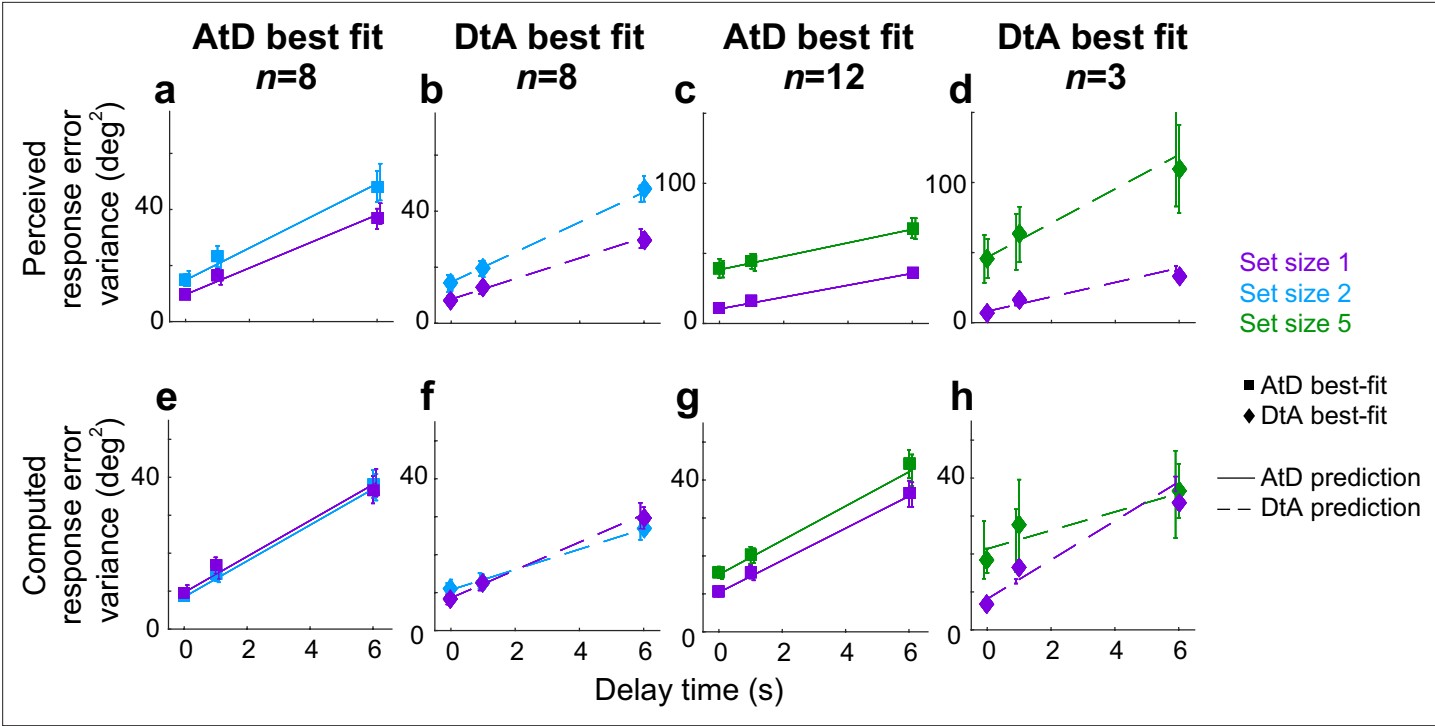

**Figure 5.** Comparison of model prediction to participant data for the Simultaneous condition. Each panel shows the empirical variance of participant errors (points and error bars are mean ± SEM data across participants) and model predictions (lines, based on the mean best-fitting parameters across participants for the given model) for the participants' best fit by the given model (AtD or DtA) for the given condition, as labeled above each column. (a–d) Perceived blocks. (e–h) Computed blocks. AtD, Average-then-Diffuse; DtA, Diffuse-then-Average.

summarized in *Figure 4c and f*, thus support the idea that for most participants, their behavior was well captured by their better-fitting model.

Summaries of the predicted report-error variances by the AtD and DtA fits for well-fit participants are shown in *Figure 5*. Overall, the model predictions qualitatively match participant behavior. In general, AtD behavior was predicted by diffusion constants that were the same for either one Perceived location or the mean Computed location based on two or five items (i.e., parallel lines in *Figure 5e and g*). DtA behavior was well predicted by diffusion constants that were larger for multiple Perceived items compared to Single Perceived items (*Figure 5f and h*). As predicted by the DtA model, the Computed errors for DtA participants were well predicted by 1/$N$th the diffusion constant for multiple Perceived items (*Figure 5e–h*). We also compared the variance in AtD and DtA participants' reports of the mean across delays using an ANOVA and multiple comparisons. For set size 2, AtD participants had a significantly higher variance in their reports than DtA participants at delay 6 ($p=0.018$), reflecting the lower effective diffusion constant of the mean for DtA than AtD when $A<1$, as was the case for our participants, on average. For set size 5, there were no significant differences in variability at any delay between models ($p>0.05$). This lack of statistical difference may reflect the low number of participants at set size 5 and/or the fact that the $A$ values of the DtA using participants at set size 5 was closer to 1, when AtD and DtA perform identically.

## Simultaneous condition strategy comparisons

Across the population, participants seemed to have different tendencies to use the two strategies (AtD or DtA) for the two set-size conditions (*Figure 6*). For set size 2, equal numbers of well-fit participants were best fit by the AtD (n=8) and the DtA (n=8), and as such neither model was significantly more likely to be a better fit across the population (Wilcoxon signed-rank test, p=0.756, *Figure 6a*, blue items). Conversely, for set size 5, more well-fit participants were better fit by the AtD (n=12) than the DtA (n=3) model (p=0.0027; *Figure 6a*, green items). Participants who were not poorly fit at either set size were more likely to be better fit by AtD in set size 5 compared to set size 2 (Wilcoxon signed-rank two-sided test for equal median log-likelihoods difference of fits of the two models across set sizes,

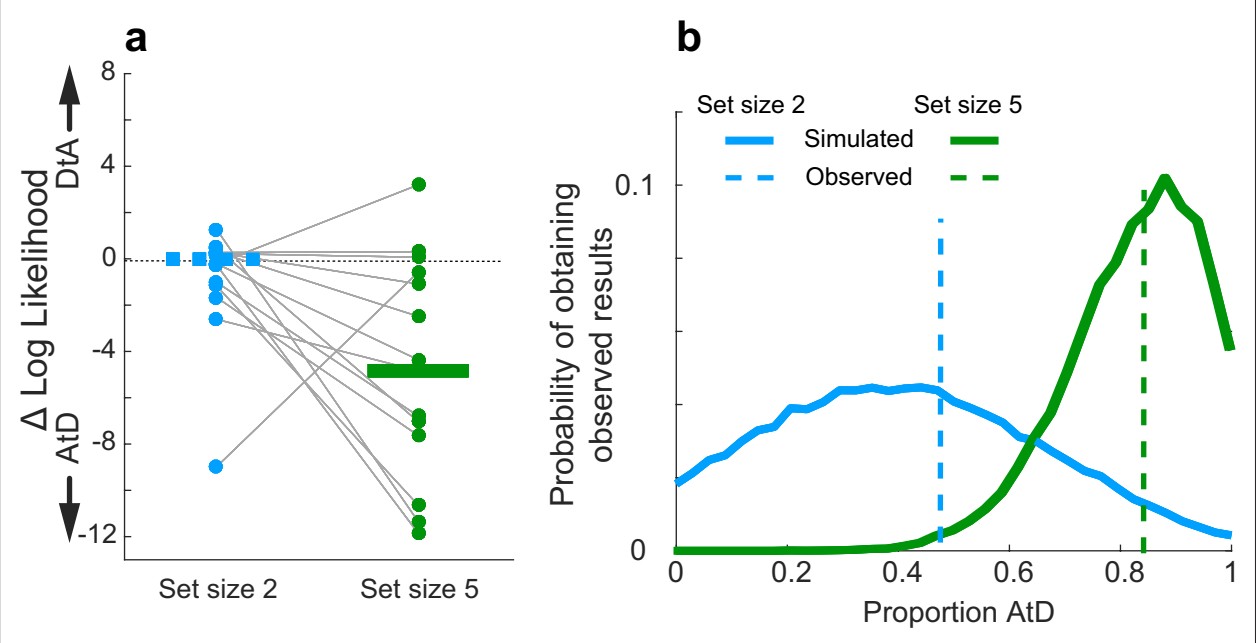

**Figure 6.** Strategy use prevalence in the population. (**a**) Difference in log-likelihood between AtD and DtA fits for the Simultaneous condition. Each point represents the difference in fit log-likelihoods for one participant; horizontal bars are medians (solid bar for set size 5 indicates two-sided Wilcoxon signed-rank test for $H_0$: median=0, p=0.0027). Positive values favor DtA, whereas negative favor AtD. Gray lines connect data generated by the same participant. Only participants whose data were well matched to one of the two models (i.e., within the 95% confidence intervals depicted in *Figure 4*) were included. (**b**) Probability of obtaining the proportion of participants' best fit by each model given average model identifiability of participant parameters. Probability of the results at set size 5 skew toward a higher proportion of AtD users compared to set size 2. AtD, Average-then-Diffuse; DtA, Diffuse-then-Average.

The online version of this article includes the following figure supplement(s) for figure 6:

**Figure supplement 1.** Relationship between log-likelihood difference for the two strategies and age for the Simultaneous condition.

p=0.029). We also found these results were robust to uncertainty associated with model identifiability (participant-wise identifiability is given in *Figure 2—figure supplement 1*). Specifically, given different possible distributions of underlying strategy prevalence (proportions), the probability of obtaining the empirically observed distributions of models shown in *Figure 6a* for either set size while considering the average model identifiability was peaked near the observed strategy proportions. This result demonstrates that the observed proportions were not likely obtained due misidentification-related chance. These probabilities distributions were also highly non-overlapping, which is consistent with a different prevalence of strategy use at the two different set sizes (*Figure 6b*).

These differences in strategy use did not correlate with the ages of the participants (Pearson correlation, *Figure 6—figure supplement 1*, p>0.20). These findings suggest that working-memory load might have affected our participants' decision strategies, such that a higher load corresponded to an increased tendency to discard information about individual samples (disk locations) and hold only the relevant computed decision variable in memory.

## Sequential condition behavior

For the Sequential condition, we separately analyzed errors for Perceived reports of disks presented at the beginning (Early) or middle (Late) of a trial. Early Perceived reports tended to be relatively unbiased (two-sided *t*-test for $H_0$: mean error=0, p>0.05; *Figure 7a*, full distributions in *Figure 7—figure supplement 1*; individual participant mean errors in *Figure 7—figure supplement 2a-d*) but became more variable over time in a roughly linear manner (*Figure 7d*), consistent with the predictions of the particle-diffusion models. For higher set sizes, errors were more variable than at lower set sizes. The rate of variance increase over time did not depend on set size (ANOVA, significant effect of set size, $F_{(2,32)}=33.44$, p=1.45e−08, and delay, $F_{(1,16)}=77.02$, p=1.64e−07, but not their interaction, $F_{(2,32)}=0.15$, p=0.256). Late Perceived reports were likewise unbiased (mean error not significantly

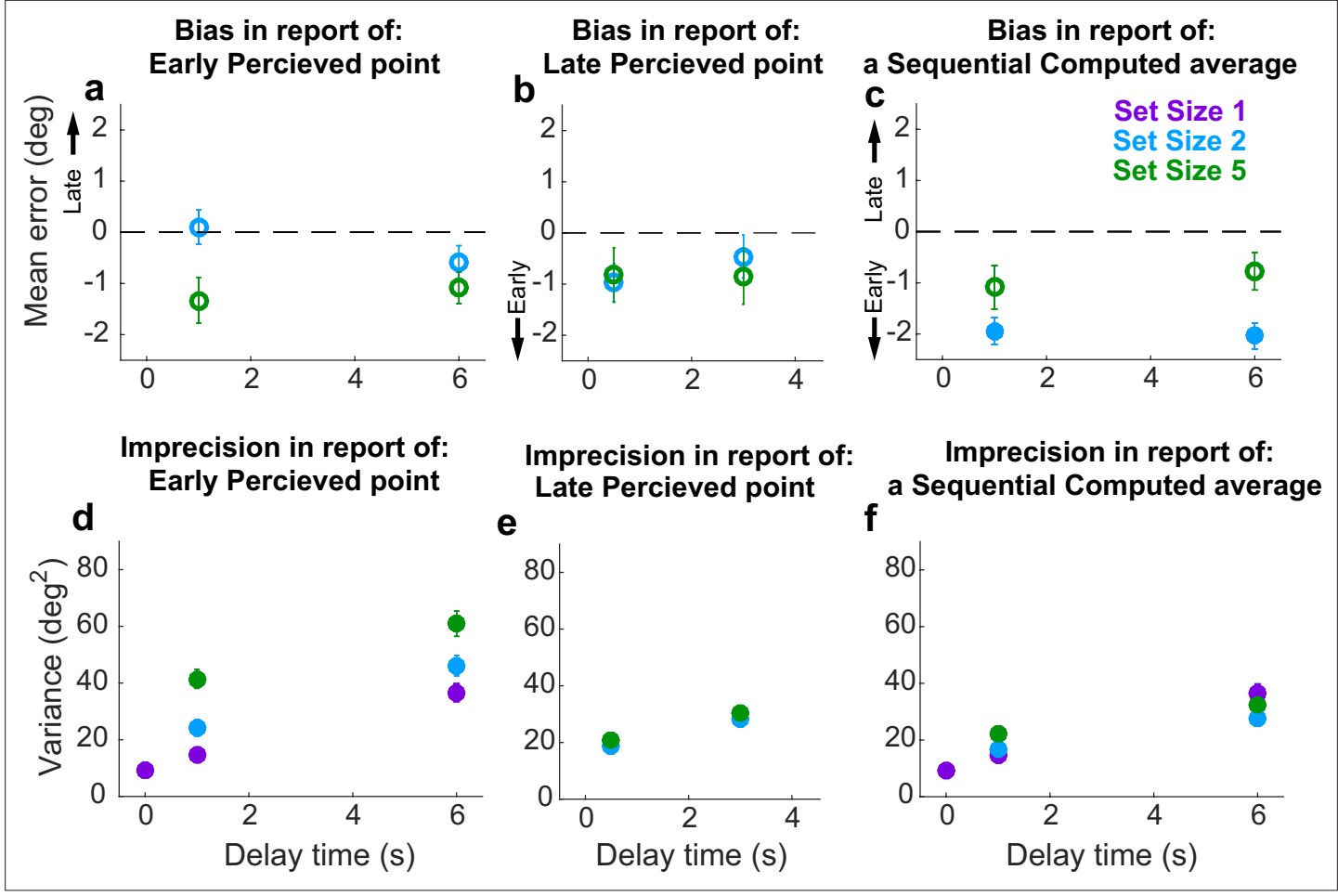

**Figure 7.** Behavioral summary for the Sequential condition. (**a**) Mean error for initially presented (Early) Perceptual items for different set sizes (colors, as indicated) and delay time (abscissa). (**b**) Mean error for midway presented (Late) Perceptual items for different set sizes (colors, as indicated) and delay time (abscissa). (**c**) Mean Computed (inferred mean) error for different set sizes (colors, as indicated) and delay time (abscissa). Filled points in (**a**–**c**) indicate two-tailed student *t*-test for $H_0$: mean=0, *p*<0.05. (**d**) Variance in Early Perceptual errors plotted as in (**a**). (**e**) Variance in Late Perceptual errors, plotted as in (**b**). (**f**) Variance in Computed (mean) errors, plotted as in (**c**). In each panel, points and error bars are mean ± SEM across participants.

The online version of this article includes the following figure supplement(s) for figure 7:

**Figure supplement 1.** Full error distributions in Sequential conditions.

**Figure supplement 2.** Participant-wise mean error in the Sequential Perceived condition.

**Figure supplement 3.** Participant-wise mean error in the Sequential Computed condition.

different from 0; *Figure 7b*, full distributions in *Figure 7—figure supplement 1*; individual participant mean errors in *Figure 7—figure supplement 2e-h*) and degraded in precision (i.e., increased in variance) over time (*Figure 7e*). However, this degradation did not depend on set size (ANOVA, significant effect of delay, $F_{(1,16)}$=39.28, *p*=1.12e−05, but not set size, $F_{(1,16)}$=0.90, *p*=0.36 or their interaction, $F_{(1,16)}$=0.0029, *p*=0.96).

Conversely, Computed (i.e., inferred mean) reports that required integrating both Early and Late items tended to be slightly biased towards the Early items for set size 2 (mean=13% of the distance between the two disks closer to the Early point than a true mean; Student's two-sided *t*-test, p<0.001) but not set size 5 (mean=3.5% closer to the mean of *N*–1 items than the true mean of all items; p>0.5; *Figure 7c*, full distributions in *Figure 7—figure supplement 1*; individual participant mean errors in *Figure 7—figure supplement 3*). The Computed report errors also increased in variance over time (*Figure 7f*). These overall errors of bias and variance did not change dramatically with training (*t*-tests for differences in mean and standard deviation of report errors between the first and second half of trials=0, p>0.05 in all cases). However, the magnitude

**Table 3.** Summary of model fits for the Sequential condition.

Parameters are: (1) $\eta_1$, non-time-dependent noise of a single value; (2) $\eta_{NE}$, non-time-dependent noise of the Early $N$–1 items; (3) $\eta_{NL}$, non-time-dependent noise of the Late $N$th items; (4) $\eta_{MN\text{-}seq}$, non-time-dependent noise of the mean of $N$ items; (5) $\sigma_1^2$, diffusion constant of a single point; and (6) A, diffusion cost of additional items. For each parameter, the maximum likelihood estimates (mean over participants ± SEM) is given for the participants' best fit with a particular model.

| | Set size (N) | Number best-fit participants | $\eta_1$ | $\eta_{NE}$ | $\eta_{NL}$ | $\eta_{MN\text{-}seq}$ | $\sigma_1^2$ | A |
|---|---|---|---|---|---|---|---|---|
| AtD | 2 | 9 | 9.44 ± 1.47 | 20.32 ± 3.89 | 16.18 ± 3.16 | 14.02 ± 1.58 | 4.44 ± 0.73 | −0.34 ± 0.44 |
| | 5 | 9 | 10.69 ± 1.21 | 37.06 ± 5.22 | 13.19 ± 2.06 | 15.94 ± 1.69 | 4.22 ± 0.73 | −0.09 ± 0.15 |
| DtA | 2 | 8 | 10.25 ± 1.53 | 18.30 ± 3.31 | 17.43 ± 1.88 | 14.00 ± 3.11 | 4.58 ± 0.52 | −3.00 ± 2.58 |
| | 5 | 8 | 9.11 ± 1.69 | 36.59 ± 5.54 | 22.27 ± 4.37 | 24.45 ± 4.60 | 4.43 ± 0.57 | −3.89 ± 2.90 |

of the errors of variance, and their change over time, depended systematically on the number of items to remember, such that more items corresponded to a slightly greater overall variance in reports at short delays, but less gain in variance over time (ANOVA, significant effect of set size, $F_{(2,32)}=7.73$, $p=1.8e{-}3$, delay, $F_{(1,16)}=73.76$, $p=2.18e{-}07$, and their interaction, $F_{(2,32)}=6.81$, $p=3.4e{-}3$). This interaction of delay and set size suggests the representation of the Computed value diffused in working memory with a different diffusion constant than for a single Perceived value. Such an interaction is consistent with predictions of both the AtD and DtA models under these conditions, though the nature of this interaction depends on the specific model, as detailed below.

## Sequential condition model fitting

To better understand the effects of delay and set size on working-memory representations of Perceived and Computed locations for individual participants under Sequential conditions, we fit the AtD and DtA models separately to data from each condition and participant (*Table 3*; the two models each had the same number of free parameters and thus were compared using the log-likelihoods of the fits). Recall that the $\eta$ parameters quantify the effect of set size on non-time-dependent noise (noise when delay is 0), whereas $\sigma_1^2$ is the model-based estimate of the diffusion constant for a single Perceived point. In general, both strategies were used by our participants in each of the two set-size conditions (for set sizes 2, 9, and 8 participants were best fit by the AtD and DtA models, respectively; for set size, 5, 8, and 8 participants were best fit by the AtD and DtA models, respectively).

Like in the Simultaneous condition, the models make identical predictions when $A=1$. Across the population, 95% CIs of $A$ did not overlap with 1, supporting the distinguishability of the two models; however, this difference from one was not always true for individual participants (Estimates of $A$ on a participant-by-participant basis are shown in *Figure 8—figure supplement 1*). For a given set size, none of the fit parameter estimates differed significantly when comparing their best-fitting values from AtD versus DtA participants (t-tests, $p>0.05$ in all cases). For participants' best fit by AtD at both set sizes, the average $A$ was close to 0, which is consistent with the lack of interaction between set size and delay seen in the Early and Late Perceptual ANOVA. Unlike in the Simultaneous condition, the participants' best fit by DtA had negative $A$ values at both set sizes, implying that the diffusion constant for multiple Perceived items became closer to 0 as the number of items increased. While counterintuitive to the concept that adding more items should increase the diffusion constant, negative $A$ values can be explained by ceiling effects: if a participant has high levels of static noise (such as in set size 5), their performance has less room to degrade while they continue to accurately track the target. As such, the rate of increase in variability $\sigma_{NE}^2$ cannot be very large and may be smaller than $\sigma_1^2$, which translates into a negative $A$ value. Alternatively, the presentation of a new item may have had a stabilizing effect on the ensemble by creating directional drift toward the new item rather than random diffusion in the remaining items (*Almeida et al., 2015*; *Wei et al., 2012*), which is not inherently accounted for in any of the present models.

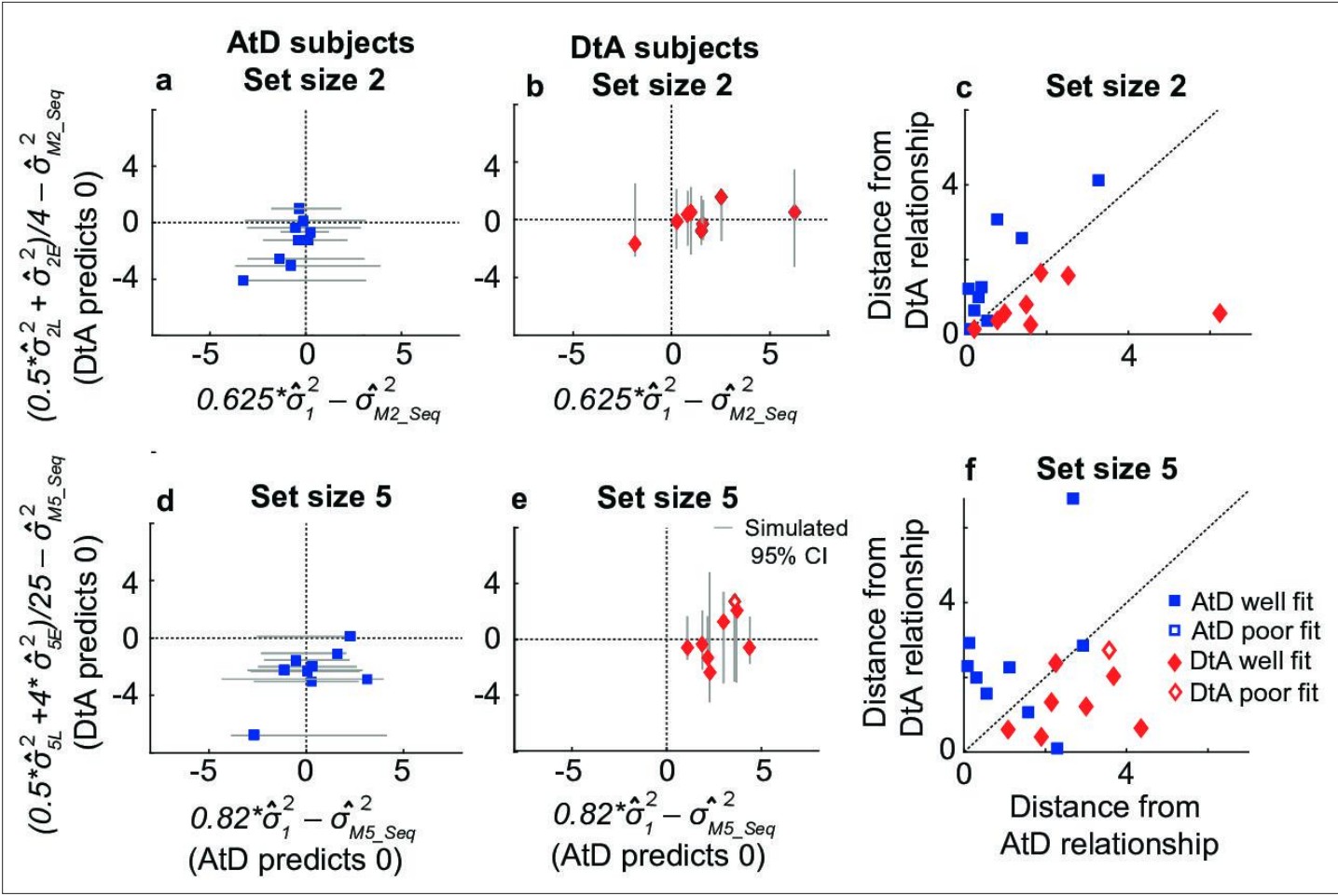

**Figure 8.** Comparisons of empirical and model-based diffusion constants. In (**a, b, d, e**), the abscissa shows the difference between: (1) empirical estimates of the diffusion constant for a Computed value measured by fitting a line to measured variance as a function of delay time for set size 2 ($\sigma_{M2}^2$, **a, b**) or 5 ($\sigma_{M5}^2$, **d, e**), and (2) the empirical estimates of the diffusion constant for a single Perceived value ($\sigma_1^2$) multiplied by the appropriate factor for the set size. The AtD model predicts a difference of 0. The ordinate shows the difference between: (1) the empirical estimate of Computed diffusion constants $\sigma_{M2}^2$ or $\sigma_{M5}^2$, and (2) the empirical estimates of the diffusion constant of a Computed value based on the DtA hypothesis. The DtA model predicts a difference of 0. Points are data from individual participants, separated by whether they were best fit by the AtD (**a, b**) or DtA (**d, e**) model for the given set-size condition. Lines are 95% confidence intervals (CIs) computed by simulating data using the best-fit parameters for the given fit and repeating the empirical estimate comparison procedure. Close symbols indicate participants who fell within the 95% CI for their best-fit model. (**e, f**) Distance of each participant's empirically estimated diffusion constant relationships from those predicted by AtD or DtA (i.e., distances from the x=0 and y=0 lines, respectively, in (**a, b, d, e**)), for set sizes 2 (**c**) and 5 (**f**). AtD, Average-then-Diffuse; DtA, Diffuse-then-Average.

The online version of this article includes the following figure supplement(s) for figure 8:

**Figure supplement 1.** Participant-specific estimates of *A* from the Sequential condition for set sizes 2 (**a**) and 5 (**b**).

## Sequential condition model validation

The Sequential condition models also make predictions about the relationship between the diffusion constants of remembered Computed and Perceived values. Once again, we assessed how well participant behavior matched these assumptions, detailed in *Equation 11* for AtD and *Equation 12* for DtA (*Figure 8*). We fit a line to the measured variances in reporting error as a function of delay for a given set size in both Perceived and Computed Sequential blocks to estimate the change in variance over time (the empirical diffusion constant estimates: $\hat{\sigma}_1^2$, $\hat{\sigma}_{NE}^2$, $\hat{\sigma}_{NL}^2$, $\hat{\sigma}_{MN\text{-}seq}^2$, where *N*=2 or 5 for the two set sizes). We then compared the difference of these empirical estimates to the predictions of the best-fit model for each participant (*Figure 8*).

In general, the participant data conformed to the model predictions of the best-fit model for each participant, despite substantial individual variability. For participants whose data were best fit by the AtD model (*n*=9 for both set sizes), the difference between empirical estimates of the diffusion

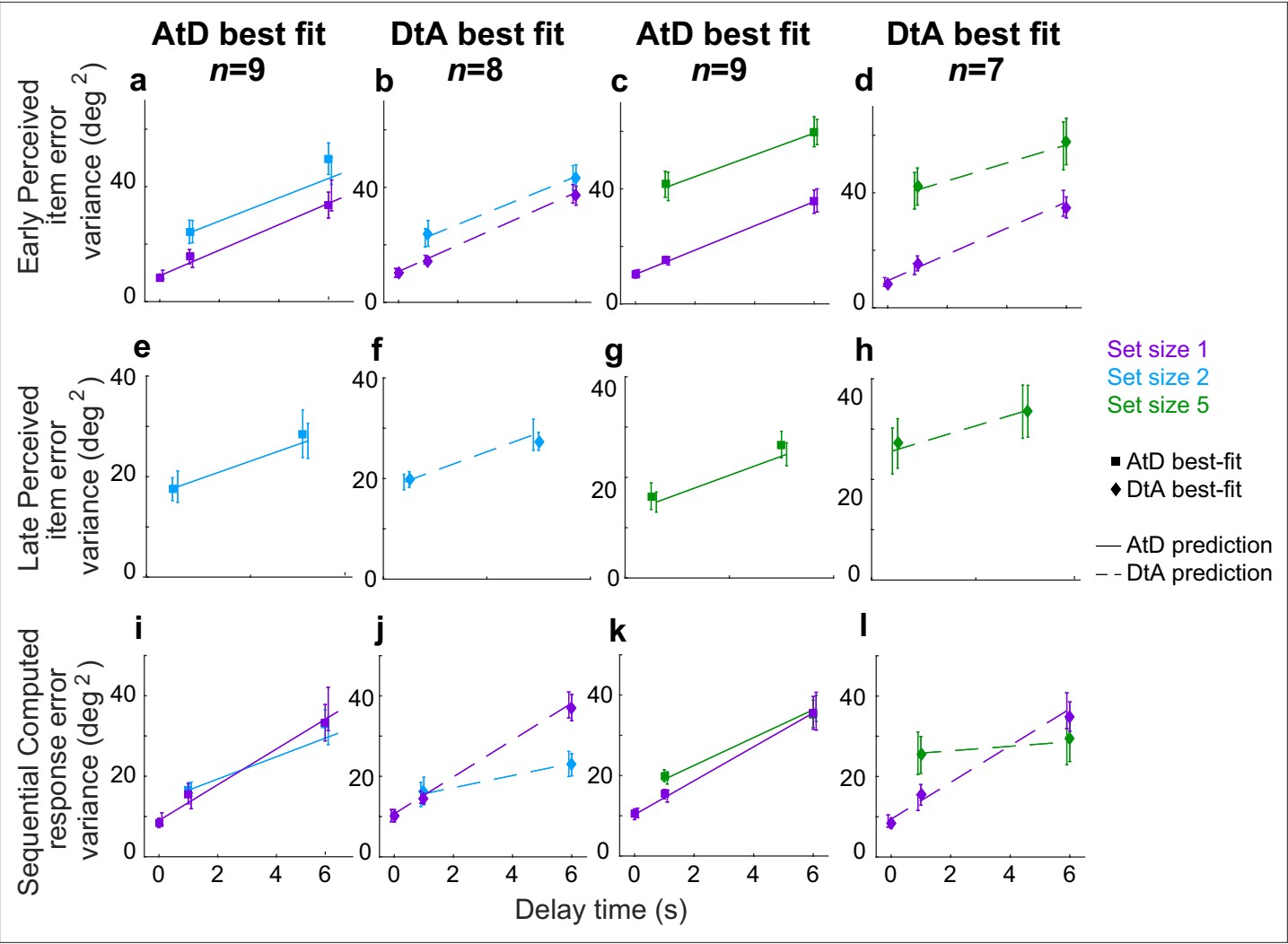

**Figure 9.** Comparison of model fits for the Sequential condition. Each panel shows the empirical variance of participant errors (points and error bars are mean ± SEM data across participants) and model predictions (lines, using mean predicted variance from each participant's best-fitting parameters for the given model) for the participants' best fit by the given model (AtD or DtA) for the given condition, as labeled above each column. (**a–d**) Errors for Ealy items in Perceived Sequential blocks. (**e–h**) Errors for Late items in Sequential Perceived blocks. (**i–l**) Errors for Sequential Computed blocks. AtD, Average-then-Diffuse; DtA, Diffuse-then-Average.

constant ($\hat{\sigma}_{MN\text{-}seq}^2$) from Computed blocks and the model-predicted equivalent fraction of the empirical estimates of the diffusion constant for a single point tended to be low (**Figure 8a and b**). Specifically, for every participant, the empirical diffusion constant differences fell within the 95% CI computed from simulations using the model fits. For participants whose data were best fit by the DtA model (n=8 for both set sizes), empirical estimates of the diffusion constant ($\hat{\sigma}_{MN}^2$) from Computed blocks tended to be similar to the expected average of empirical estimates of the diffusion constant for multiple items (0.5 $\hat{\sigma}_{NL}^2+(N–1)*\hat{\sigma}_{NE}^2)/N^2$; **Figure 8d and e**. Specifically, for seven participants, empirical diffusion constant differences fell within the 95% CI computed from simulations using the model fits. The remaining participant was considered poorly fit and not considered in further analyses. These analyses, which are summarized in **Figure 8c and f**, thus support the idea that for most participants, their behavior was well captured by their better-fitting model.

Summaries of the predictions of report errors variances for AtD and DtA fits are shown in **Figure 9**. In general, participants' best fit by AtD exhibited diffusion constants that were, on average, lower for Computed than Perceived values (**Figure 9i and k**; lower slope of cyan/blue line vs. purple line). This difference decreased with increased set size, which is expected from the averaging process (**Figure 2d**). Additionally, both the Early and Late variances were, on average, fairly well matched by

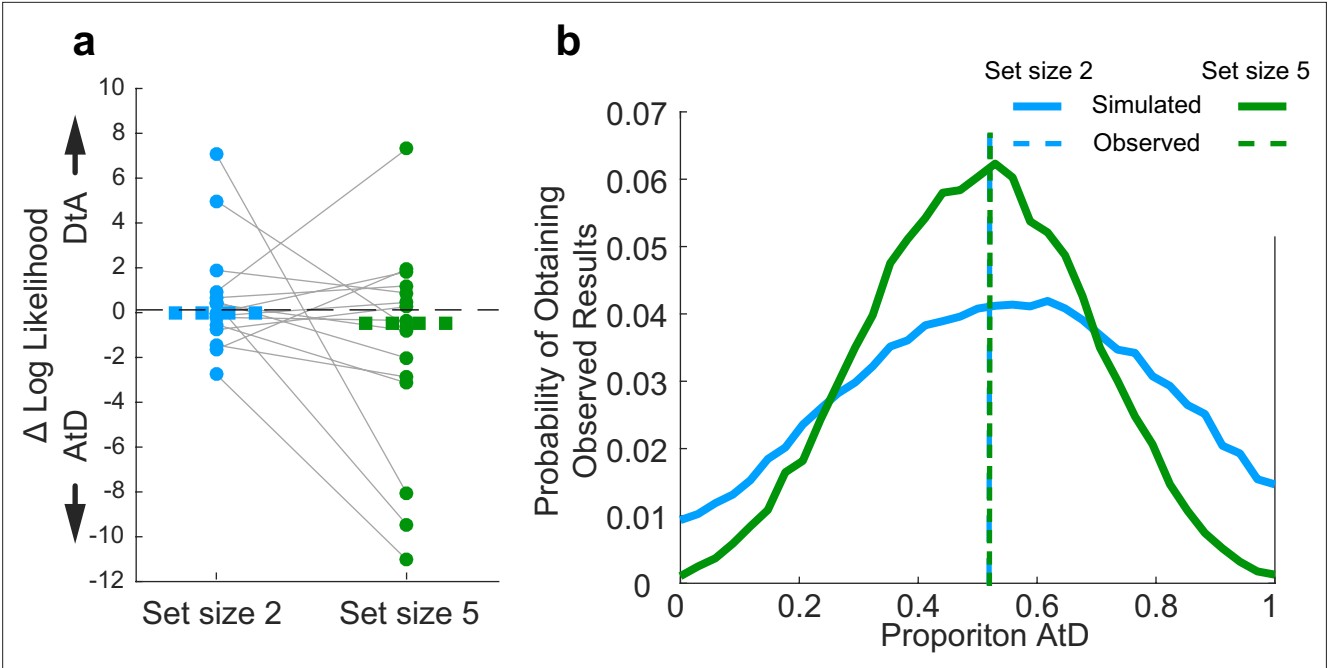

**Figure 10.** Assessment of strategy use prevalence in the population in Sequential conditions. (**a**). Difference in log-likelihood per well-fit participant AtD and DtA fits. Negative values favor AtD. Each point represents the difference in fit log-likelihoods for one participant and data from the same participant are connected across set sizes; horizontal bars are medians. Positive values favor DtA, whereas negative values favor AtD. We failed to reject the null hypothesis (two-sided Wilcoxon signed rank test for $H_0$: median=0, $p>0.05$) for both set sizes. (**b**). Probability of obtaining the proportion of participants' best fit by each model given average model identifiability of each participant parameters. Probability of the results at set sizes 2 and 5 are most likely when the probability of AtD and DtA are similar. AtD, Average-then-Diffuse; DtA, Diffuse-then-Average.

The online version of this article includes the following figure supplement(s) for figure 10:

**Figure supplement 1.** Relationship between log-likelihood difference for the two strategies and age for the Sequential condition.

their model predictions (*Figure 9a, e, c and g*). Conversely, participants' best fit by DtA exhibited diffusion constants that were notably smaller for Computed mean locations versus single Perceived locations (*Figure 9j and l*; lower slope of cyan/blue line vs. purple line). The corresponding average predictions by the best fit DtA models for error variance of Early and Late items also aligned with participant data from DtA fit participants (*Figure 9b, f, d and h*). We also compared the variance in AtD and DtA participants' reports of the mean across delays using an ANOVA and multiple comparisons but found no significant differences in variability at any delay between models ($p>0.05$).

## Sequential condition strategy comparisons

Across the population, participants had roughly equal tendencies to use either one of the two strategies (AtD or DtA) for the two set-size conditions (*Figure 10*). For set size 2, one more participant was best fit by the AtD (*n*=9) versus the DtA (*n*=8) model (Wilcoxon signed-rank two-sided test for the median difference in the log-likelihoods of fits of the two models to data from each participant=0, $p=0.868$). For set size 5, two more participants were best fit by the AtD (*n*=9) versus the DtA (*n*=7) model ($p=0.234$). Participants well fit at either set size were not significantly more likely to be fit by either model across set sizes (Wilcoxon signed-rank two-sided test for identical median log-likelihoods difference of fits of the two models across set size, $p=0.283$). Given different possible distributions of underlying strategy prevalence (proportions), the probability of obtaining the empirically observed distributions of models shown in *Figure 10a* for either set size while considering the average model identifiability was peaked near the observed strategy proportions. This result demonstrates that the observed proportions were not likely obtained due to misidentification-related chance (*Figure 10b*). These differences in strategy use did not correlate with age of participants (*Figure 10—figure supplement 1*, Pearson correlation, $p>0.20$). Thus, on average, participants lost fidelity in their representations

of a Computed value when it needed to be computed from sequentially presented information, as in many processes of evidence accumulation. The dynamics of this degradation differed for the two strategies, neither of which was statistically more likely than the other across our participants.

## Alternative models for Sequential conditions

Up to this point, we have considered the two extremes of either: (1) holding all stimuli in memory until the time of the report (the DtA model), or (2) averaging all stimuli as soon as possible and then holding only this single average in memory (the AtD model). When set size is larger than 2, it is possible to perform a hybrid of these strategies. For example, one could diffuse the initial $N$–1 items until the final stimulus is presented and then combine all evidence at that point. Thus, we did an additional log-likelihood comparison using this hybrid model for set size 5 (note that AtD and this hybrid model are identical for set size 2, and all models had the same number of parameters). We found that three participants previously identified as AtD and three participants previously identified as DtA were slightly better described by the hybrid model (average LL difference for formerly DtA participants=0.34, average difference from formerly AtD participants 1.28). This finding is consistent with the idea that people likely use a spectrum of strategies, including nuanced combinations of AtD-like and DtA-like dynamics (which were the focus of the present study, to demonstrate how those dynamics can give rise to identifiable signatures of behavioral errors) that warrant further consideration when building more detailed models of memory-dependent decision-making.

## Strategy comparisons across conditions

The use of different strategies (i.e., those captured by the AtD and DtA models) did not appear to reflect a tendency of individual participants to use a particular strategy across different conditions. Specifically, we used Fisher's exact test of independence based on set size across temporal conditions as well as based on temporal conditions across set sizes to test whether individual participants were best fit by the same model under different task conditions. We failed to reject the null hypothesis that there is no relationship between a participant's strategy use across set size for both Simultaneous and Sequential conditions (i.e., strategy use in set size 2 Simultaneous was not predictive of use in set size 5 Simultaneous, nor was it for Sequential conditions; $p=0.31$ and $p=1$, respectively). We also failed to reject the null hypothesis that there is no relationship between a participant's strategy use across temporal conditions for both set sizes 2 and 5 ($p=0.54$ and $p=1$, respectively). Thus, we found that only under set size 5 were Simultaneous conditions participants more likely to use one strategy (AtD) over the other (DtA). In all other tested cases, participants were equally likely to use either strategy, and strategy use was not predictive across conditions for individual participants.

# Discussion

The goal of this study was to better understand if and how capacity and temporal limitations of working memory affect human decision-making. We used a task that required participants to report remembered spatial locations based on different numbers of objects and for different delay durations, both of which are known to systematically affect the precision of memory reports (*Bastos et al., 2018*; *Cowan et al., 2008*; *Funahashi et al., 1989*; *Oberauer et al., 2016*; *Panichello et al., 2019*; *Ploner et al., 1998*; *Schneegans and Bays, 2018*; *White et al., 1994*). We used two pairs of conditions to investigate these effects across decision-making circumstances. The first condition was Perceptual versus Computed, which allowed us to recapitulate previous findings of the effects of capacity and temporal limitations of working memory for directly observed (perceptual) quantities and then extend those findings to the kind of computed quantity that is used as a decision variable for tasks that require integration or averaging to reduce uncertainty (*Brody and Hanks, 2016*; *Gold and Shadlen, 2007*; *Ratcliff et al., 2016*; *Shadlen and Shohamy, 2016*; *Summerfield and Tsetsos, 2012*). The second was Simultaneous versus Sequential conditions, which extended our investigation to include the effects of working-memory limitations on decision-making under relatively simple conditions (i.e., when all relevant evidence was presented at once) to the effects in a basic case of evidence accumulation over time (i.e., in which a new piece of evidence is used to update a computed quantity).

Our primary finding was that computed variables based on either simultaneously or sequentially presented information were susceptible to the same kinds of working-memory constraints as

perceived variables. These working-memory limitations corresponded to a decrease in precision over time, which places critical constraints on the kinds of decision variables that are required to persist over time, such as when decisions are delayed. Specifically, the variability caused by singular events such as sensory encoding or averaging tended to be ~10° in Simultaneous conditions, whereas noise accumulating during memory delay periods was typically ~3.5°/s. Therefore, after ~3 s, the noise that can be attributed to memory versus non-memory sources is about equal, and after this point memory noise begins to dominate overall variability. This result appears to contradict previous findings that found no effect of extra delays on the effectiveness of evidence accumulation for certain decisions (*Liu et al., 2015*; *Waskom and Kiani, 2018*). However, those studies used tasks with binary choices that required decision variables with less clear sensitivity to the kinds of working-memory effects we found in the context of a continuous, spatially based decision variable. Additionally, we found that increasing the number of decision-relevant items also decreased the accuracy of the continuous decision variable, although the nature of this effect was variable. More work is needed to fully characterize the conditions under which temporal and capacity limitations on the precision of working-memory representations affect decisions based on those representations.

We also found that the exact nature of interactions between working-memory limitations and decision-making depend critically on the strategy used to form the decision, and those strategies can vary substantially across individuals and tasks. For our tasks, we focused on two primary strategies. The first strategy, captured by the AtD model, stipulated that a participant first calculates and then stores the Computed value. Its key prediction is that a Computed value should be susceptible to the same effects of working-memory limitations as a single remembered Perceptual value in simultaneous conditions. The second strategy, captured by the DtA model, stipulated that all individual values are stored in working memory until the time of decision. Its key prediction is that the overall rate of variance increase is inversely related to the number of items. Although the differences in performance of adopters of each of the strategies were often minimal in the present study, the differences in strategy rely on rates of degradation over time and thus performance differences would be expected to grow over longer delays. We found that participants tended to use an AtD strategy for the Simultaneous conditions with a relatively high load (five items), but otherwise were roughly equally likely to use either strategy, including for all Sequential conditions.

This finding of multiple strategy use raises several intriguing future questions. For example, we found that for the Simultaneous condition, several individuals switched from using DtA for the smaller set size to AtD for the larger set size, but we do not know if this switch was a consequence of their personal working-memory capacities. From an optimality standpoint, DtA better preserves a computed value compared to AtD for a given level of non-time-dependent noise and cost per storage item (*A*), but only if *A* remains low (<1). It would be interesting to see if for more intermediate set sizes (i.e., three or four items) there is a reliable increase in the probability of a participant using AtD with a progression that relates to other measures of the individual's working-memory capacity. Such future studies would more definitively support the conclusion that increased working-memory load corresponds to an increased tendency to discard information about individual samples and hold only the computed decision variable in memory. Future studies should also examine other factors that might govern which strategy is used for a given set of conditions. For example, participants in our study were instructed to report the average but given no additional details about how to do so, nor given strong incentives for choosing any particular strategy versus another. Future studies could provide more detailed instructions, incentives, and/or feedback to better understand the flexibility with which these different strategies can be employed.

Future work should also examine in more detail several other facets of working memory that were not included in our models but in principle could affect decision variables that are computed and retained over time. First, we did not consider possible differences in metabolic energy and other resources needed to implement the different working-memory demands of different strategies (*van den Berg and Ma, 2018*). Future studies of strategy heterogeneity may need to consider how different strategies minimize both response errors and execution costs. Second, our DtA model assumed no interference between multiple items stored in memory. This assumption is undoubtedly an oversimplification, given that storage of multiple items has been both hypothesized and shown to create attraction and repulsion (*Almeida et al., 2015*; *Krishnan et al., 2018*; *Wei et al., 2012*). Such directional drift can create a decrease in variance over time that could affect decision variables that involve

multiple quantities stored at once. Third, our DtA model also assumed that each item was stored individually. Alternatively, items could have been discarded or merged (chunked) (*Krishnan et al., 2018*; *Wei et al., 2012*), leading to different memory loads which could also affect performance. Fourth, most of our participants used strategies that were well described by the AtD or DtA model. However, under certain conditions (i.e., Sequential, set size 5) some participants seemed to use hybrid strategies. This kind of strategy would suggest extensive flexibility in when and how evidence is incorporated into computed decision variables, thereby placing potentially complex demands on working memory.

Both of our primary models were based on assumptions of a drifting memory representation. This random drift is traditionally associated with attractor models of working memory (*Bays, 2014*; *Compte et al., 2000*; *Macoveanu et al., 2007*; *Wei et al., 2012*) that have been used extensively to describe the underlying neural mechanisms (*Funahashi et al., 1989*; *Shafi et al., 2007*; *Takeda and Funahashi, 2002*; *Wimmer et al., 2014*). In these models, neural network activity is induced by an external stimulus and then maintained via excitatory connections of similarly tuned neurons and long-ranged inhibition. Random noise causes the center of this activity (which represents the stimulus) to drift in a manner that, dependent on the implementation, can depend on the delay duration, set size, and/or their interaction (*Almeida et al., 2015*; *Bays, 2014*; *Koyluoglu et al., 2017*). A recent implementation even can naturally compute a running average based on sequentially presented information (*Esnaola-Acebes et al., 2021*). Our results imply that such models should be extended to support the flexible use of different strategies that govern when and how incoming information is used to form such averages. It will be interesting to see if such a flexible model can account for neural activity in the dorsolateral prefrontal cortex, which includes neurons with persistent activity that has been associated with both spatial working memory (*Compte et al., 2000*; *Constantinidis et al., 2018*; *Riley and Constantinidis, 2016*; *Wei et al., 2012*; *Wimmer et al., 2014*) and the formation of decisions based on an accumulation of evidence (*Curtis and D'Esposito, 2003*; *Heekeren et al., 2006*; *Heekeren et al., 2008*; *Kim and Shadlen, 1999*; *Lin et al., 2020*; *Philiastides et al., 2011*).

In conclusion, we found that in this spatial, continuous task, participant accuracy for both perceived and computed values was subject to working-memory limitations of both time and capacity. Additionally, we found behavior that was consistent with both the storage strategies we investigated. The fact that different participants employed different strategies for storing a computed value (such as a decision variable) and that these strategies have different consequences on overall accuracy has important implications for not only future neural network models of working memory, but also for future computational models of decision-making.

# Materials and methods
## Human psychophysics behavioral task
We tested 17 participants (4 males, 12 females, 1 chose not to answer; age range=22–87 years). The task was created with PsychoPy3 (*Peirce et al., 2019*) and distributed to participants via Pavlovia.org, which allowed participants to perform the task on their home computers after providing informed consent. These protocols were reviewed by the University of Pennsylvania Institutional Review Board (IRB) and determined to meet eligibility criteria for IRB review exemption authorized by 45 CFR 46.104, category 2.

Participants were instructed to sit one arm-length away from their computer screens during the experiment and to use the mouse to indicate choices. Each participant completed 1–2 sets of four blocks of trials in their own time.

The basic trial structure is illustrated in *Figure 1*. Each trial began with the presentation of a central white fixation cross (1% of the screen height). The participant was instructed to maintain fixation on this cross when not actively responding. The participant began each trial by placing the mouse over the cross and clicking, to allow for self-pacing and pseudo-fixation. Initiating a trial caused a white annulus of radius 25% of the screen height to appear. A block-specific memory array appeared 250 ms later, centered at an angle chosen uniformly and at random on the annulus. The array consisted of 1, 2, or 5 colored disks sized 1.5% screen in diameter. The angular difference between any two adjacent disks was at least 6°, and between the two most distal disks was at most 60°. The disks from clockwise to counter-clockwise were always presented in the same order: green, red, blue, magenta,

and yellow. When fewer than five disks were presented, the latter colors were omitted. The consistent color ordering was intended to reduce errors caused by misbinding of location and color. The angular differences between disks in an array were randomly selected from five preselected sets of five angular differences that obeyed the restrictions stated above, centered on a randomly selected location on the circle. If set size was <5, later numbers were omitted. The sets were [–22, –11, –2, 7, 13], [–25,–4, 6, 12, 24], [–30,–18, 3, 15, 29], [–22,–10, 0, 7, 17], and [–19,–12, 0, 9, 28] (numbers are degrees clockwise relative to the randomly selected location on the circle).

The memory array remained on the screen for 0.5 s, while the annulus remained on the screen throughout the delay of 0, 1, or 6 s. At the end of the delay, the fixation cross was replaced with a response cue that either matched a color of a disk in the memory array, indicating a response to the remembered location of that disk, or was white, indicating a response to the mean angle of all disks in the present trial. The response type varied by block (see below). The participant then moved the mouse and clicked on the annulus at a position at which they remembered the requested response. Feedback was then given indicating the correct location, the participant's response, and the difference between the two.

We used four block-wise conditions: (1) Simultaneous Perceived blocks used arrays of 1, 2, or 5 disks presented simultaneously at the beginning of the trial. Participants were told in advance that they would always be asked to report the location of one of the array disks but were not informed which one until the response period. The probed disk was picked randomly on each trial. (2) Simultaneous Computed blocks used arrays of 2 or 5 disks presented simultaneously at the beginning of the trial. Participants were told in advance they would need to report the average angle of all disks shown in the present trial. (3) Sequential Perceived blocks were identical to Simultaneous Perceived blocks, except only arrays of 2 or 5 disks were used, and all but one of the disks (the counter-clockwise most) was presented at the beginning of the trial. The final disk was presented for 0.5 s ending midway through the delay of 1 or 6 s. The most counter-clockwise disk was always the last presented disk, to make the task easier. Participants were told in advance that the final disk would be presented in the middle of the delay for these blocks. (4) Sequential Computed blocks were identical to Simultaneous Computed blocks, but with delayed presentation of the final disk as in Sequential Perceived blocks. Again, participants were told in advance that the final disk would be presented in the middle of the delay.

All participants completed one and most (12) participants completed two blocks of each type. Each block contained 50 trials at each set size and each delay time, the order of which was randomized.

## Basic analyses

Trials were excluded from analysis if the response was >30° from the correct angle. This cutoff was based on assessment of the error distributions (*Figure 3—figure supplement 1*, *Figure 7—figure supplement 1*); using a cutoff of 25° did not noticeably change the results. On average, <10% of trials were excluded per delay condition per set size per block (see *Figure 3—figure supplement 1*, *Figure 7—figure supplement 1*). These trials were excluded to focus analysis on trials that were directed toward the correct location and avoid lapses of attention and extreme motor errors. We investigated both the bias and variance in participant responses, as follows.

We quantified bias as the mean error between the response and the true probed angle for each participant and condition (positive/negative values imply errors that were systematically counter-clockwise/clockwise, respectively). A Bonferroni-corrected two-sided *t*-test was used to assess whether this mean response error was significantly different from zero across participants for each set size, delay, response type, and temporal presentation. Additionally, the mean error and CI for each participant were calculated for each condition (*Figure 3—figure supplements 2 and 3*; *Figure 7—figure supplements 2 and 3*). For the Sequential condition, we also assessed how bias Computed responses were compared to the true mean location. We took the difference of the reported mean from the *N*th point and normalized this difference by the distance between the mean of *N*–1 items and the *N*th item. For set size 2, the true mean had a normalized value of 0.5. For set size 5, the true mean had a normalized value of 0.8.

We quantified the variance of the error between the response and the true probed angle for each participant and condition. We chose variance as opposed to other measures of dispersion for consistency with our particle models (see below) in which variance scales linearly with delay. We examined

the effects of set size, delay duration, and task context on response variability using a two-way repeated measures ANOVA. On Simultaneous Perceived and Computed blocks, we used a 3 (delay duration: 0, 1, or 6 s) × 3 (set size: 1, 2, or 5 disks) within-participant design. On Sequential Perceived blocks, we used a 2 (delay duration: 1 or 6 s) × 3 (set size: 1, 2, or 5 disks) within-participants design for stimuli presented at the beginning of the trial (Early) and a 2 (delay: 0.5 or 3 s) × 2 (set size: 2 or 5 disks) design for stimuli presented halfway through the trial (Late). On Sequential Computed blocks, we used a 2 (delay duration: 1 or 6 s) × 3 (set size: 1, 2, or 5 disks) within-participants design. When the comparison included set size=1, data were always taken from the Simultaneous Perceived block.

To assess performance differences based on strategy use, additional analyses were performed once the data had been fit to the models and the best fit model had been selected (see below). These analyses included an assessment of response error variability in the Computed blocks using a 2 (model: AtD or DtA) × 3 or 2 (delay: 0, 2, or 6 s Simultaneous condition, 1 or 6 s for Sequential) ANOVA with multiple comparisons to identify differences. To interrogate best fit parameter differences, two-sided *t*-tests were used to see if the mean difference in best-fit parameter between AtD and DtA participants was significantly different from 0 for both Simultaneous and Sequential conditions. To assess learning effects, a two-sided, paired *t*-test was used to see if the mean or standard deviation of error responses in set size 5 Sequential conditions differed between the first and second half of trials (we found no difference at either delay: for 1 s delay p=0.67 and 0.11 for mean and standard deviation, respectively; for 6 s delay p=0.75 and 0.98 for mean and standard deviation, respectively).

## Model-based analyses

Our models were based on principles of working memory that are well described by bump-attractor network models (**Compte et al., 2000**; **Laing and Chow, 2001**; **Wimmer et al., 2014**). In such models, stimulus location is represented by a 'bump' in activity from neurons tuned to that and similar locations. These neurons recurrently activate each other, maintaining a bump of activity even after stimulus cessation. However, because of the stochastic nature of neural activity and synaptic transmission (**Faisal et al., 2008**), there is variability in which neurons have the most activity at any given time (and thus are the center of the bump representing the stimulus). This variability in bump center corresponds to variability in the location representation and a degradation of the memory representation over time. The dynamics of this bump can be described as a diffusion process that obeys Brownian motion (**Compte et al., 2000**; **Kilpatrick, 2018**; **Kilpatrick et al., 2013**; **Laing and Chow, 2001**). We used this simplified description in our models as follows.

## Perceived values in working memory

A single point (i.e., the central spatial location of a single disk), $x_1$, is assumed to be represented in working memory by $x_{t,1}$, where $t$ represents the time since the removal of the stimulus. We assume that $x_{t,1}$ evolves like a sample from a Brownian-motion process. Specifically, when $x_1$ is observed, it is encoded with some perceptual noise, $\eta^p$. Therefore, at time zero, $x_{0,1} \sim N(x, \eta^p)$. This representation accumulates noise over time with some diffusion constant, $\sigma_1^2$, further degrading the representation of $x_{t,1}$ from $x_1$ such that $x_{t,1} \sim N(x_1, \eta^p+t*\sigma_1^2)$. There is additional motor noise in the participant's report, $r_{t,1}$, and we denote the variance of this motor noise by $\eta^m$. Mathematically, it is equivalent to add the motor noise at the beginning or the end of the diffusion of $x_{t,1}$ when considering the report, $r_{t,1}$. In our model, we thus represent the sum of the perceptual and motor noise as a single, static noise term. Hence, we show simulated trajectories of $x_{t,1}$ in *Figure 2a* with an initial variance of $\eta_1=\eta^p+ \eta^m$, so that at time $t$, the report $r_{t,1}$ is the current angle of the trajectory. Therefore

$$r_{t,1} \sim N(x_1, \eta_1 + t^*\sigma_1^2). \tag{1}$$

We used Gaussian rather than von Mises distributions because: (1) they are easier to generalize to other, non-circular domains; (2) the Gaussian standard deviation parameter has a more intuitive interpretation than the von Mises concentration parameter; and (3) our stimuli were constrained to be a maximum of 60° apart, and thus the periodicity of the von Mises distribution was unnecessary to capture the diffusion dynamics.

When multiple items are held in memory, they are held with less fidelity than a single point (**Bays et al., 2009**; **Brady and Alvarez, 2015**; **Koyluoglu et al., 2017**; **Wei et al., 2012**). We therefore assume that the sum of the initial perceptual noise and final motor noise, with variance denoted by $\eta_N$,

can depend on the number of disks, $N$. Moreover, we describe, $x_{t,n}$, the representation of the $n$th item at time $t$, by a normal distribution with a diffusion constant that is potentially higher than for a single point. We assume that this new diffusion constant $\sigma_N^2$, equals $\sigma_1^2 * N^A$ and thus scales as a power of the total number of stimuli, $N$, held in memory (**Bays et al., 2009**; **Bays and Husain, 2008**; **Wei et al., 2012**), and is proportional to the diffusion constant corresponding to a single stimulus representation, $\sigma_1^2$. Therefore:

$$r_{t,n} \sim N(x_n, \eta_N + t^* \sigma_1^{2*} N^A) \tag{2}$$

All representations in a set of size $N$ share the same magnitude of non-time-dependent noise, $\eta_N$, but the evolution of each representation is assumed to be independent. To examine distributions of responses across the various presented locations, we measured the error of the response $r_{t,n}$ relative to the true location of the target the observer was asked to report, $x_{t,n}$. According to our model, the difference between the true and reported location (the error, $e_{t,n}$) is

$$e_{t,1} \sim N(0, \eta_1 + t^* \sigma_1^2) \tag{3a}$$

$$e_{t,n} \sim N(0, \eta_N + t^* \sigma_1^2 {}^* N^A) \tag{3b}$$

The linear relationship between total accumulated noise and time for both a single and multiple memoranda is illustrated in **Figure 2b**.

## AtD simultaneous model

For this model, the representation of the average is stored as a single particle that diffuses the same as a Perceived item (i.e., a location at which there was a visible stimulus; see **Figure 2b**). Thus, the diffusion term for the representation of a computed average of $N$ items, $\sigma_{MN}^2$, is also $\sigma_1^2$. We do not assume that the representation of the average has the same static noise as a single point, because there could be additional noise from inaccurately averaging multiple items or conversely a reduction in overall noise resulting from the averaging of multiple random variables (the constituent items). We denote the variance of the static noise for the Computed mean by $\eta_{MN}$. The difference between the true mean of $N$ stimuli and the mean reported at time $t$ is therefore:

$$e_{t,mN,AtD} \sim N(0, \eta_{MNAtD} + t^* \sigma_1^2) \tag{4}$$

## DtA simultaneous model

For this model, the individual perceived items are stored as individual, independently diffusing particles and then averaged at the end of the trial. Thus, the diffusion constant of the Computed value is the variance of the average of $N$ random variables each with the diffusion constant $\sigma_1^2 * N^A$, resulting in an effective diffusion constant for the Computed value of $\sigma_{MN}^2 = \sigma_1^2 * N^A / N$, where the division by $N$ arises from averaging. Again, we allow for a free non-time-dependent-noise term because of the uncertain effects of the averaging calculation itself. For this model, the error in the reported location at time $t$ of the average of the mean, $M$, of $N$ items, $e_{t,MN}$, is:

$$e_{t,mN,DtA} \sim N(0, \eta_{MNDtA} + t^* \sigma_1^2 {}^* N^A / N) \tag{5}$$

If $A=1$, the AtD and DtA models are identical. We thus used best-fitting values of $A$ to help assess model distinguishability for each participant and task condition (see **Figure 2—figure supplement 1**; **Figure 4—figure supplement 1**). If $A<1$, then the DtA strategy results in a lower diffusion constant for a Computed value than predicted by the AtD model and results in a smaller average reporting error (see **Figure 2c**). If $A>1$, then AtD results in the lower diffusion constant and thus a lower average reporting error. However, given the parameter estimates obtained in this study, we did not find that participants necessarily used the strategy that would result in the lowest response variability.

## Sequentially presented values in working memory

In the Sequential blocks, $N–1$ items were presented immediately (Early items), and the $N$th item was presented halfway through the delay (Late point). Therefore, both models assume that the diffusion constant for the representation of the $N–1$ early items increases with the addition of the $N$th item,

corresponding to the increased memory load. The representation of the Late item then diffuses for only half of the delay time, $T$ (see **Figure 2d and e**). We formalized this process with the following model for the report error of the Early ($e_{T,NE}$) and Late ($e_{T,NL}$) items:

$$e_{T,NE} \sim N(0, \eta_{NE} + T/2^*\sigma_1^2{}^*(N-1)^A + T/2^*\sigma_1^2{}^*N^A) \tag{6a}$$

$$e_{T,NL} \sim N(0, \eta_{NL} + T/2^*\sigma_1^2{}^*N^A) \tag{6b}$$

Here, $T$ is the total time of the delay, and we assumed different non-time-dependent noise for both Early and Late items ($\eta_{NE}$ and $\eta_{NL}$, respectively).

## AtD sequential model

This model assumes that the Early items are averaged immediately and stored as a single item. At $t=T/2$, the Late item is presented and combined immediately, through appropriate weighted averaging, with the mean of the Early items. This new mean again diffuses with the same accumulating noise as a single item (see **Figure 2d**). Therefore:

$$e_{T,MN-seqAtD} \sim N(0, \eta_{MN-seqAtD} + ((N-1)/N)^2{}^*T/2^*\sigma_1^2 + T/2^*\sigma_1^2) \tag{7}$$

At $t=T/2$, the representation of the $N$th item has not accumulated any diffusion noise and only has non-time-dependent noise, which is absorbed in the $\eta_{MN\text{-}Seq}$ term. The first time-dependent term, $((N-1)/N)^2{}^*T/2^*\sigma_1^2$, results from the appropriate weighted averaging of the mean of the Early items (time-dependent noise of $T/2^*\sigma_1^2$) with the Late item (time-dependent noise=0). The final term, $T/2^*\sigma_1^2$, is the diffusion of the resultant mean until the end of the delay.

## DtA sequential model

This model assumes that the representations of all $N$ items diffuse as they are presented, resulting in $N–1$ items described by **Equation 6a** and one item described by **Equation 6b**. These items are then averaged at the end of the delay, resulting in an overall error of:

$$\begin{aligned} e_{T,MN-seqDtA} \sim N(0, \eta_{MN-seqDtA} + (T/2^*\sigma_1^2{}^*N^A + ... \\ (N-1)[T/2^*\sigma_1^2{}^*(N-1)^A + T/2^*\sigma_1^2{}^*N^A])/N^2) \end{aligned} \tag{8}$$

where the constant noise terms from the Early and Late items are absorbed in the $\eta_{MN\text{-}SeqDtA}$ term, the next term $T/2^*\sigma_1^2{}^*N$. is the diffusion in the representation of the last disk shown, and the remaining terms arise from the first $N–1$ items shown. The effect of this averaging on the effective diffusion constant is shown in **Figure 2e**.

## Alternative Model

A third model was considered for Sequential set size five conditions: the first $N–1$ items diffused until the $N$th point was presented, at which point all items were averaged and this average diffused for the remainder of the delay. Thus, $e_{T,MN-seqDtA} \sim N(\eta_{MN\_SeqAlt} + (N-1)^*(T/2^*\sigma_1^2{}^*(N-1)^A)/N^2 + T/2^*\sigma_1^2)$.

## Model fitting

The AtD and DtA models were fit to data from the Simultaneous Perceived and Computed blocks using five free parameters: (1) the static noise of a single point ($\eta_1$), (2) the diffusion noise of a single point ($\sigma_1^2$), (3) the static noise of $N$ items ($\eta_N$), (4) the exponent of storing $N$ items ($A$), and (5) the static noise of the mean ($\eta_{MN(AtD or DtA)}$). We fit these models for $N=2$ and $N=5$ conditions separately, using trials from the following conditions. Perceived: delays 1, 3, and 6 s; array sizes 1 (**Equation 3a**) and $N$ (**Equation 3b**). Computed: delays 1, 3, and 6 s; array size $N$ (**Equation 4** for AtD, **Equation 5** for DtA). To validate the assumption that the cost of storing additional items ($A$) was constant between Perceived and Computed blocks for DtA fit participants, we refit the models using trials from only Perceived or only Computed trials. The difference in the best fit $A$ values were compared across participants using a two-sided $t$-test for mean difference=0.

The AtD and DtA models were fit to data from the Sequential Perceived and Computed blocks using six free parameters. The additional parameter accounted for differences in the static noise for Early and Late items. We fit these models for $N=2$ and $N=5$ conditions separately, using trials from the

following conditions. Perceived: delays 1, 3, and 6 s; array size 1 (*Equation 3a*). Perceived: delays 3 and 6 s, array size $N$ for both Early (*Equation 6a*) and Late (*Equation 6b*) items. Computed: delays 3 and 6 s, array size $N$ (*Equation 7* for AtD or *Equation 8* for DtA).

Because the mean error for each individual participant was not always 0, when fitting the AtD and DtA models we used the empirical mean error from the condition being fitted as a fixed bias term in the model. Mean error and CIs for each participant for each condition are shown in *Figure 3—figure supplements 2 and 3*; *Figure 3—figure supplements 2 and 3*.

We obtained separate maximum-likelihood fits for AtD and DtA models for each individual participant, using the function fmincon in MATLAB to minimize the summed negative log-likelihood of obtaining the observed errors for a given condition according to the above equations. Initial parameter values were randomized and the fitting repeated to avoid local minima. Because all models within a given condition had the same number of parameters, we compared log-likelihoods to determine the best-fitting model for a given participant. Because the number of parameters is the same, comparing likelihoods produces equivalent model selection to BIC or AIC.

## Assessing model assumption and identifiability

To assess how well each participant's data matched the assumptions of the AtD and DtA models, we also fit a line to the variances of response errors across delays for a given condition for a given participant to obtain empirical estimates of the various diffusion constants (e.g., slope of lines in *Figure 2b*; empirical estimate of a Perceived value, $\sigma_N^2$, for set size $N$; empirical estimate of a Computed value, $\sigma_{MN}^2$, set size $N$). These empirical estimates of the diffusion constants did not enforce the relationships imposed by the AtD and DtA models between the different diffusion constants in each model, respectively. We compared the relationships of these empirical estimates of diffusion constants to the relationships assumed by our models, as follows:

AtD Simultaneous. The Computed mean diffuses with the same diffusion constant as a single value. Thus:

$$\sigma_1^2 - \sigma_{MN}^2 = 0 \tag{9}$$

DtA Simultaneous. The Computed mean is the average of $N$ items each diffusing with a constant of $\sigma_N^2$. Thus:

$$\sigma_N^2/N - \sigma_{MN}^2 = 0 \tag{10}$$

AtD Sequential. The time-dependent noise has variance that increases as $((N–1)/N)^2*T/2*\sigma_1^2+T/2*\sigma_1^2$ (*Equation 7*). Factoring out $T$ gives the diffusion constant for the Computed mean, $\sigma_{MN}^2=[(N–1)^2+N^2]/(2\,N^2)*\sigma_1^2$. Thus:

$$[(N-1)^2 + N^2]/(2N^2)^* \sigma_1^2 - \sigma_{MN}^2 = 0 \tag{11}$$

DtA Sequential. The time-dependent noise has variance that increases as $T/2*\sigma_1^2*N^A+(N–1)[\,T/2*\sigma_1^2*(N–1)^A+T/2*\sigma_1^2*N^A])/N^2$ (*Equation 8*). By *Equation 6a*, the diffusion constant for an Early Perceived point, $\sigma_{NE}^2$, is $[0.5*\sigma_1^2*(N–1)^A+0.5*\sigma_1^2*N^A]$ and by *Equation 6b*, the diffusion constant for a Late Perceived point, $\sigma_{NL}^2$, is $\sigma_1^2*N^A$. Factoring out $T$ and substituting gives the diffusion constant for the Computed mean, $\sigma_{MN}^2=(0.5\sigma_{NL}^2+(N–1)*\sigma_{NE}^2)/N^2$. Thus:

$$(0.5^*\sigma_{NL}^2 + (N - 1)^* \sigma_{NE}^2)/N^2 - \sigma_{MN}^2 = 0 \tag{12}$$

To assess how well the relationships between participant empirical estimates of the diffusion constants matched these assumptions, for each participant, we simulated 1000 iterations of a participant performing the task using the best-fit model for the given true participant and the maximum-likelihood estimate parameters for that participant. We then estimated the empirical diffusion constants for each of these iterations in the same way that we did for our participants, namely by fitting a line to the measured variance of the simulated errors across delays, for each condition and iteration. Our 1000 simulations gave us an expected range around the expected diffusion constant relationships detailed in *Equations 9–12* to compare to our participants' empirical diffusion constant

relationships. Participants whose empirical diffusion constant relationships fell within the central 95% of the simulated expected range were considered well fit by their model.

To assess model identifiability, for each participant and condition, we fit both models to the results of each set of 1000 simulations generated using the best-fitting parameters from the best-fitting model for that participant and condition. We used the log-likelihoods to determine the best model for each simulation and determined the percentage of correctly identified models. We used these results to further assess the reliability of our analyses of strategy prevalence across the population of participants, as follows. For each condition (Simultaneous vs. Sequential and set size 2 vs. 5), we determined an empirical proportion of AtD versus DtA prevalence (i.e., in terms of the relative numbers of participants whose data were best fit by the two models). We then sampled from a binomial distribution 10,000 times using a range of possible proportions between 0 and 1 in 1/34th increments. Each iteration yielded a simulated proportion, which we then adjusted with our model-identifiability results: each simulated AtD participant had a chance of being misidentified according to average percent correctly identified for that model as determined above, and likewise with DtA. The proportion of samples that matched our data was used to create the probability of obtaining our observed results, given possible underlying proportions and average model identifiability (*Figures 6b and 10b*).

## Acknowledgements

The authors thank Adrian Radillo and Gaia Tavoni for their discussions early in the development of this project, particularly regarding early formulations of the models used here and the task structure.

## Additional information

### Competing interests

Joshua I Gold: Senior editor, eLife. The other authors declare that no competing interests exist.

### Funding

| Funder | Grant reference number | Author |
| --- | --- | --- |
| National Institute of Mental Health | R01 MH115557 | Kresimir Josic<br>Zachary P Kilpatrick<br>Joshua I Gold |

The funders had no role in study design, data collection and interpretation, or the decision to submit the work for publication.

### Author contributions

Kyra Schapiro, Conceptualization, Formal analysis, Investigation, Methodology, Project administration, Software, Visualization, Writing – original draft, Writing – review and editing; Krešimir Josić, Zachary P Kilpatrick, Supervision, Writing – review and editing; Joshua I Gold, Funding acquisition, Project administration, Supervision, Writing – review and editing

### Author ORCIDs

Kyra Schapiro http://orcid.org/0000-0001-8308-0744
Zachary P Kilpatrick http://orcid.org/0000-0002-2835-9416

### Ethics

Human subjects: The task was created with PsychoPy3 and distributed to participants via Pavlovia.com, which allowed participants to perform the task on their home computers after providing informed consent. These protocols were reviewed by the University of Pennsylvania Institutional Review Board (IRB) and determined to meet eligibility criteria for IRB review exemption authorized by 45 CFR 46.104, category 2.

### Decision letter and Author response

Decision letter https://doi.org/10.7554/eLife.73610.sa1
Author response https://doi.org/10.7554/eLife.73610.sa2

## Additional files

### Supplementary files
• Transparent reporting form

### Data availability
All analysis code is available on GitHub (https://github.com/TheGoldLab/Memory_Diffusion_Task, copy archived at swh:1:rev:69cee7449f92f9d19148332979087bf4e6a9f867). Data used for figures will be made available on Dryad.

The following dataset was generated:

| Author(s) | Year | Dataset title | Dataset URL | Database and Identifier |
|---|---|---|---|---|
| Schapiro K, Josic K, Gold J, Kilpatrick Z | 2022 | Memory array locations, delay times, and participant response | https://doi.org/10.5061/dryad.w3r2280rm | Dryad Digital Repository, 10.5061/dryad.w3r2280rm |

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
