## [Editor Report]

This paper employs sophisticated modeling of human behavior in well-controlled tasks to study how limitations of working memory constrain decision-making. Because both are key cognitive processes, that have so far largely been studied in isolation, the paper will be of broad interest to neuroscientists and psychologists. The observed working memory limitations support previous findings and extend them in critical ways.

---

## [Decision Letter]

**Decision letter after peer review:**

Thank you for submitting your article "Strategy-dependent effects of working-memory limitations on human perceptual decision-making" for consideration by *eLife*. Your article has been reviewed by 2 peer reviewers, and the evaluation has been overseen by Tobias Donner as the Reviewing Editor and Michael Frank as the Senior Editor. The following individual involved in review of your submission has agreed to reveal their identity: Peter R Murphy (Reviewer #1).

Essential revisions:

1) Alternative strategies for sequential task.

The authors claim that some subjects follow the AtD strategy and others the DtA strategy but experimental evidence for this claim seems weak. Take Figure 10 as an example (Figure 6 is similar). The authors conclude from the data in Figure 10 that on the population level there is no significant difference between the models. On an individual subject level, the δ_LL_ values are small (for most subjects | δ_LL_ | < 3) which one could interpret as either model fitting the data equally well.

In order to claim that there are indeed two different strategies in place, it needs to be shown that the data can only be explained by heterogeneous strategies (for example following a methodology as in Stephan et al. Neurimage 2009 and Rigoux et al. Neuroimage 2014).

Regarding the sequential task: It may be worth considering a mixed strategy model as an alternative because it may explain the data better. Specifically, subjects would follow the DtA strategy until the last stimulus is observed and then switch to the AtD strategy until the end of the delay (i.e., compute the average in the middle of the trial, once all the evidence has been observed).

2) Appropriateness of modeling choices.

The A parameter, governing the relationship between the diffusion constant for a single point and the constants for multiple points, seems estimated differently in AtD and DtA models: in AtD, it's estimated using only data from Perceived blocks with set size > 1, and it plays no role in the AtD process (only, instead, in the memory maintenance process during the delay period of Perceived trials); whereas in DtA, it's estimate using data from both the same Perceived blocks, ‘and’ the Compute blocks at equivalent set sizes. This raises two concerns.

i. Because A parameters in each model are effectively fit to different data, any comparison of the parameter estimates (which is invited by placing them in same table and by some of the discussion in the text [p.9]) needs to be carefully qualified in the associated text.

ii. There is an implicit assumption that the A parameter is fixed across Perceived and Computed blocks. However, Perceived trials with set size > 1 require working memory maintenance of a ‘conjunction’ of stimulus features (location and colour), whereas the latter require maintenance (assuming the DtA process is employed) of only a single feature per stimulus (location); thus, it can reasonably be expected that the effect of load may be more severe in Perceived than Computed blocks. It seems that this possibility is not allowed for in the presented model fits.

Recommendations:

a. The above concerns could be addressed by fitting another round of models, this time fitting A in the DtA model using data from only Computed blocks.

b. In addition, A estimates should be compared between fits of the AtD and DtA models (something that is not possible given the fits as currently presented): if there is a systematic difference between the two, this would indicate that A is indeed different in Perceived and Computed blocks and this should be accounted for in the fits.

3) Implications of model fits.

The implications of strategy choice could be further illuminated by examining what factors if any (overall accuracy of judgments; magnitude of non-time-dependent model parameters) differentiate AtD and DtA adopters. Further clarification of what differentiates working memory from decision computations on the presented tasks could be achieved by addressing the following questions through further analysis and/or discussion: How should the decision-specific (eta_MN_) parameters be interpreted in the context of other prominent models of decision-making? How does their magnitude compare to other noise sources? Does this speak to the question of whether the predominant source of noise in decision-making is sensory-/motor- or memory-related or related the decision computation itself?

4) Clarity of presentation.

The Results section is difficult read, and several key aspects of the approach and findings are only clarified during careful reading of the Methods section. Most prominently, there is insufficient explanation of key model predictions that may be counterintuitive for many readers; a lack of clarity around what individual model parameters capture; and confusing elements to how the model fits are presented. We encourage the authors to carefully revise the Results section with this concern in mind.

Specific recommendations:

a. Implications of AtD vs DtA strategy choice:

The fact that, all else being equal, the DtA strategy generates ‘more’ precise behaviour on Computed trials than the AtD model (at least for the parameter range human participants seem to occupy here) is the central feature that differentiates behaviour produced by the two strategies and renders the models identifiable. The authors also seem to take the direction of this effect to be self-evident, as no effort is made to explain to the reader why this pattern emerges. For instance, readers may wonder whether allowing for N = [2, 5] sources of gaussian noise compared to only 1 source should actually produce ‘more’ variability in behaviour. Now, the averaging over particles that takes place at the culmination of the DtA process counteracts the greater total noise to produce less variability in behavioural reports. But this was far from self-evident, and this key effect should be unpacked.

b. Model parameter descriptions:

There seems to be a general lack of clarity around what exactly each model parameter, and in particular different subscripts to different parameters, are supposed to capture. In Figure 2, for example, subscripts N, MN, 1, N(E/L) and MNSeq are all used but only explained in Methods.

c. Alternative strategy interpretation. (see also point 3):

Please clarify what exactly the finding is, because this currently seems ambiguous: Compare line 244 "(…) participants had roughly equal tendencies to use either of the two strategies" implying that we can distinguish which strategy individual subjects are following, vs. line 257 "(…) neither of which was more likely than the other for a given participant" which implies the opposite.

d. Model fitting procedures (see also point 2):

The role of the A parameter in the different fits is confusing, specifically, seeing fits of A for the AtD model since, this parameter does not seem to used at all in the AtD process. If our understanding is correct, the A parameter in these fits is only relevant to producing behavior in Perceived blocks with set size > 1 – a condition of the experiment to which the AtD process is never actually applied. But at no point is this made explicit, leaving room for quite considerable confusion when the reader encounters this important section of the Results.

[Editors' note: further revisions were suggested prior to acceptance, as described below.]

Thank you for resubmitting your work entitled "Strategy-dependent effects of working-memory limitations on human perceptual decision-making" for further consideration by eLife. Your revised article has been evaluated by Michael Frank (Senior Editor) and a Reviewing Editor.

The reviewers have discussed their reviews with one another, and the Reviewing Editor has drafted this to help you prepare a revised submission. Both reviewers were overall very positive about your revisions and felt that the manuscript is much more accessible now. Both support publication in eLife.

Essential revisions:

1) Please address one outstanding clarification question by Reviewer #1, with which Reviewer #2 agreed. Rather than summarizing, we paste the original reviewer point below. Once this point is addressed, the paper can be accepted without additional review.

I only have one lingering point of confusion that I would welcome clarification on. This again centres around treatment of the A parameter in the AtD model. The authors write in the current manuscript (p.7) that in this model "the average is calculated immediately upon observing the evidence and then stored as a single particle in working memory" (lines 95-96) and then "the single estimate held in working memory diffuses with the same diffusion constant as a single perceived item (σMN^2^ = σ1^2^) (lines 97-99). Based on this my understanding is that there is only ever one particle diffusing in the AtD model during Computed blocks, regardless of set size; this particle always has the same diffusion constant (σ1^2^), and there is, therefore, ‘no role’ for set size/the A parameter in determining diffusion noise during Computed-block memory maintenance in the AtD model. Why, then, is it later written that "Because of the previously described relationships between σ1^2^, σN^2^, and σMN^2^ it is therefore also true that in the AtD model σN^2^ = σMN^2^ * N^A^" (lines 109-110)? Given the earlier sentences, the only way I can see this being true is if N here refers to the number of ‘particles’ being maintained in memory (which, in the AtD model, is always equal to 1, and so the N^A^ term is doing no work here and just causes considerable confusion) – and not the set size presented to the participant, as N is consistently used to denote elsewhere in the paper. I'm sorry if I'm missing something here, but this seems a key conceptual point to get right for clear presentation and differentiation between the two models. The new Table 1, and my careful reading of the Methods, seems consistent with my own intuition that A plays no role on Computed blocks in the AtD model. But this seems fundamentally inconsistent with the equation emphasized on lines 109-110; and indeed with the authors' response to point 2 in the first round of reviews, which I must confess I did not understand.

Now, assuming my own interpretation is correct, and that indeed the A parameter is not doing any work on Computed blocks in the AtD model (instead, in this model it only serves to set the diffusion noise across different set sizes in ‘Perceived’ blocks), then I stand by my initial point that without clarification, it is misleading to include and invite comparison of fitted A parameters for the AtD and DtA models in the same table (new Table 2). In one case (AtD) the A parameter only captures (and in turn, will only be constrained by) behaviour in Perceived blocks; in the other (DtA) it captures (and is constrained by) behaviour in both Perceived and Computed blocks. But currently, this is never made explicit.

*Reviewer #1 (Recommendations for the authors):*

I thank the authors for engaging well with all comments and suggestions from the first round of reviews. In my opinion the new draft – including more detailed explanation of the models and their predictions early on – is a lot more accessible. I also found the new model identifiability analyses to be quite convincing, in the sense that they provide further evidence to support the claim the distinct strategies are indeed being used by different participants and that while the specifically identified proportions are subject to quite some uncertainty (especially for low set sizes), this key result is nonetheless recoverable given the data at the authors' disposal. Altogether, these additions reaffirm my initial impression that this manuscript is a valuable contribution to the field, breaking new ground in connecting working memory and decision-making.

I only have one lingering point of confusion that I would welcome clarification on. This again centres around treatment of the A parameter in the AtD model. The authors write in the current manuscript (p.7) that in this model "the average is calculated immediately upon observing the evidence and then stored as a single particle in working memory" (lines 95-96) and then "the single estimate held in working memory diffuses with the same diffusion constant as a single perceived item (σMN^2^ = σ1^2^) (lines 97-99). Based on this my understanding is that there is only ever one particle diffusing in the AtD model during Computed blocks, regardless of set size; this particle always has the same diffusion constant (σ1^2^), and there is, therefore, ‘no role’ for set size/the A parameter in determining diffusion noise during Computed-block memory maintenance in the AtD model. Why, then, is it later written that "Because of the previously described relationships between σ1^2^, σN^2^, and σMN^2^ it is therefore also true that in the AtD model σN^2^ = σMN^2^ * N^A^" (lines 109-110)? Given the earlier sentences, the only way I can see this being true is if N here refers to the number of ‘particles’ being maintained in memory (which, in the AtD model, is always equal to 1, and so the N^A^ term is doing no work here and just causes considerable confusion) – and not the set size presented to the participant, as N is consistently used to denote elsewhere in the paper. I'm sorry if I'm missing something here, but this seems a key conceptual point to get right for clear presentation and differentiation between the two models. The new Table 1, and my careful reading of the Methods, seems consistent with my own intuition that A plays no role on Computed blocks in the AtD model. But this seems fundamentally inconsistent with the equation emphasized on lines 109-110; and indeed with the authors' response to point 2 in the first round of reviews, which I must confess I did not understand.

Now, assuming my own interpretation is correct, and that indeed the A parameter is not doing any work on Computed blocks in the AtD model (instead, in this model it only serves to set the diffusion noise across different set sizes in ‘Perceived’ blocks), then I stand by my initial point that without clarification, it is misleading to include and invite comparison of fitted A parameters for the AtD and DtA models in the same table (new Table 2). In one case (AtD) the A parameter only captures (and in turn, will only be constrained by) behaviour in Perceived blocks; in the other (DtA) it captures (and is constrained by) behaviour in both Perceived and Computed blocks. But currently, this is never made explicit.

*Reviewer #2 (Recommendations for the authors):*

The authors have followed the recommendations of the previous decision letter and the additional analysis confirm the findings of the first version. I have no further issues.

---

## [Author Response]

Essential revisions:1) Alternative strategies for sequential task.The authors claim that some subjects follow the AtD strategy and others the DtA strategy but experimental evidence for this claim seems weak. Take Figure 10 as an example (Figure 6 is similar). The authors conclude from the data in Figure 10 that on the population level there is no significant difference between the models. On an individual subject level, the δ_LL_ values are small (for most subjects | δ_LL_ | < 3) which one could interpret as either model fitting the data equally well.In order to claim that there are indeed two different strategies in place, it needs to be shown that the data can only be explained by heterogeneous strategies (for example following a methodology as in Stephan et al. Neurimage 2009 and Rigoux et al. Neuroimage 2014).

We thank the reviewers for their suggestion of using Variable Bayesian Analysis (VBA) to assess conclusions regarding heterogeneous strategy use in the population. We investigated the use of this analysis by running our log likelihoods through the VBA toolbox. For the Simultaneous set size 5 and both Sequential data sets, the VBA agreed with our conclusions: of a greater frequency of AtD use in set size 5 and heterogeneous populations for both sequential conditions. However, for Simultaneous set size 2, the VBA suggested that there was a significantly greater frequency of AtD subjects than DtA subjects, in contrast to our finding that 8 subjects were best described by AtD and 9 by DtA (though one was not considered well fit). To try to better understand this discrepancy, we simulated 100 rounds of task performance using the parameters and models that were best fit to data from our subject population. Thus, for this simulated data set we knew the true underlying distribution of parameters and models. We then ran the VBA on data generated from these simulations. For our set size 2 Simultaneous simulations, we found that the VBA consistently overestimated the proportion of AtD subjects compared to the ground truth. We include a summary of these results in Author response image 1. We concluded that VBA was not an effective way to reliably recover population distributions and therefore chose not to include these analyses in our revised manuscript.

**Author response image 1. sa2fig1:** Results of VBA simulations compared to known ground truth. a. For the set size 2, Sequential condition, VBA results tended to agree with the ground truth that neither strategy was more frequent (62% of simulations were within 10% of the ground truth). b. For the set size 2, Simultaneous condition, VBA results tended to disagree with the ground truth (only 23% of simulations were within 10% of the ground truth). c. For the set size 5, Simultaneous condition, VBA results tended to agree with the ground truth (90% of simulations were within 10% of the ground truth).

Instead, we further analyzed the identifiability of our models (percent of simulations correctly identified by LLR) and the probability of obtaining the observed model use distributions of our participants. We quantified model identifiability as the percent of simulated data for which the correct model had a lower negative log likelihood than the alternative. We now report these results for Simultaneous conditions in Figure 2 —figure supplement 1. We then used these identifiability results to run simulations of different underlying model-use proportions. For 10000 iterations of each sampled underlying model-use proportion (using values that ranged between 0 and 1 in 1/34 steps), we sampled 17 subjects and assigned them each a model (AtD or DtA) according to the given model-use proportion. We then used our model-identifiability results to reverse the model assignments according to our estimated probability of misidentifying each model. We used these percentages to calculate the probability of obtaining our results for different “true” underlying proportions of model prevalence given average model identifiability to supplement our log likelihood difference results. The results, presented in Figures 6b and 10b, support our conclusions that: (1) for the Simultaneous condition, more subjects used AtD for set size 5 versus set size 2; and (2) for the Sequential condition, roughly equal numbers of subjects used each strategy type for the two set sizes. The probability functions are also peaked near the underlying observed distributions, indicating that average model identifiability was sufficiently high to prevent the observed participant distribution from being observed by random chance due to misidentification.

Regarding the sequential task: It may be worth considering a mixed strategy model as an alternative because it may explain the data better. Specifically, subjects would follow the DtA strategy until the last stimulus is observed and then switch to the AtD strategy until the end of the delay (i.e., compute the average in the middle of the trial, once all the evidence has been observed).

We have considered the reviewer’s thoughtful suggestion of a DtA model that transitions to AtD when the final stimulus is observed. For set size 2, this suggested model is identical to AtD (because there is only one point to diffuse until the last stimulus is observed) and was thus not investigated further. In contrast, for set size 5, the suggested model is distinct, if not quite as separable from the existing two models. We include a new section (lines 353–367) reporting that when this model is included in the model selection process, 3 former AtD subjects and 3 former DtA subjects are (slightly) better fit by this hybrid. This result suggests that for more complex decision-making scenarios, a more nuanced spectrum of strategies may be in use.

2) Appropriateness of modeling choices.The A parameter -- governing the relationship between the diffusion constant for a single point and the constants for multiple points -- seems estimated differently in AtD and DtA models: in AtD, it's estimated using only data from Perceived blocks with set size > 1, and it plays no role in the AtD process (only, instead, in the memory maintenance process during the delay period of Perceived trials); whereas in DtA, it's estimate using data from both the same Perceived blocks, ‘and’ the Compute blocks at equivalent set sizes. This raises two concerns.

We regret the confusion and have clarified in the revised manuscript that the *A* parameter, which describes the relationship between the diffusion constants for Perceived set size 1 and set size *N*, is fit to the same data for the two models: each time we fit the parameter using trials of all possible delays from Perceived set size 1, set size *N*, and Computed set size *N* (revised manuscript lines 155-156).

To further clarify, the main difference between the two models is that: 1) the AtD model enforces relationships between the diffusion constants for Perceived set size 1 and Computed set size *N*, and thus the *A* parameter also describes the relationship between the diffusion constants for Perceived set size *N* and Computed set size *N*; whereas 2) the DtA model enforces relationships between the diffusion constants for Perceived set size *N* and Computed set size *N*, and thus the *A* parameter also describes the relationship between the diffusion constants for Perceived set size 1 and [Computed set size *N*]’*N* (because the Computed diffusion constant is defined in DtA as 1/*N*^th^ the Diffusion constant for Perceived set size *N*) (lines 107-109). Accordingly, estimates of the *A* parameter are obtained using all of the aforementioned trials.

i. Because A parameters in each model are effectively fit to different data, any comparison of the parameter estimates (which is invited by placing them in same table and by some of the discussion in the text [p.9]) needs to be carefully qualified in the associated text.

See above; we clarified in the text that the *A* parameters for each model are fit to the same data.

ii. There is an implicit assumption that the A parameter is fixed across Perceived and Computed blocks. However, Perceived trials with set size > 1 require working memory maintenance of a ‘conjunction’ of stimulus features (location and colour), whereas the latter require maintenance (assuming the DtA process is employed) of only a single feature per stimulus (location); thus, it can reasonably be expected that the effect of load may be more severe in Perceived than Computed blocks. It seems that this possibility is not allowed for in the presented model fits.

See below for responses to the specific recommendations.

Recommendations:a. The above concerns could be addressed by fitting another round of models, this time fitting A in the DtA model using data from only Computed blocks.

We thank the reviewers for this excellent suggestion. We tested and report in the revised manuscript (lines 164–169) that there is no statistically significant difference when comparing best-fitting values of *A* estimated from Perceived blocks alone versus Computed blocks alone for DtA participants. Therefore, there is no evidence that the memory load (as captured by *A*) was affected by the requirements of conjunction in the Perceived vs the Computed blocks.

We also now note (Figure 1 legend, Methods: 508–509) that we used a consistent ordering of the colors of the disks to minimize any extra memory load caused by the conjunction of location and color and balance memory load between Perceived and Computed blocks (as measured by *A*).

b. In addition, A estimates should be compared between fits of the AtD and DtA models (something that is not possible given the fits as currently presented): if there is a systematic difference between the two, this would indicate that A is indeed different in Perceived and Computed blocks and this should be accounted for in the fits.

We now report (Table 2, Table 3) that best-fitting values of *A* were slightly higher for DtA versus AtD subjects, although the difference was statistically significant (*p*<0.05) only for set size=5 Simultaneous. It is certainly true that our models do not capture all aspects of each participant’s strategy, but we still believe that the distinction of DtA versus AtD strategies parameterized in this way provides a parsimonious account of the primary strategy classes that they used.

3) Implications of model fits.The implications of strategy choice could be further illuminated by examining what factors if any (overall accuracy of judgments; magnitude of non-time-dependent model parameters) differentiate AtD and DtA adopters. Further clarification of what differentiates working memory from decision computations on the presented tasks could be achieved by addressing the following questions through further analysis and/or discussion: How should the decision-specific (eta_MN_) parameters be interpreted in the context of other prominent models of decision-making? How does their magnitude compare to other noise sources? Does this speak to the question of whether the predominant source of noise in decision-making is sensory-/motor- or memory-related or related the decision computation itself?

We thank the reviewers for this excellent suggestion. We now include an additional analysis of the factors that differentiate performance and parameters of AtD and DtA adopters. In brief, we found a significant performance difference in terms of the variability of responses for the average of AtD and DtA adopters at set size 2, delay 6, simultaneous conditions. These comparisons are now reported in lines: 172–174 and Table 2 for Simultaneous parameters, 211–216 for Simultaneous variability over conditions, 286–287 and Table 3 for sequential parameters, and 332–334 for sequential variability over conditions.

Regarding the broader questions about the decision-making/sensory/motor/memory related noise sources, we do not feel that our current study design is best suited to address these concerns. Eta_MN_ represents all static (i.e., not diffusion-dependent) noise sources that affected the participants’ decisions. This term thus captures a combination of initial sensory cue location, motor, and averaging (decision) noise. The individual contribution of these different noise sources is difficult to disentangle given our study design. We found that eta_MN_ was typically ~10°, whereas σ^2^_MN_ was ~3–4 °/second. As a consequence, after ~3 sec the majority of the response variability can be attributed to diffusion rather than the other factors, which we now note in the discussion (lines 403–407). Other studies, such as Drugowitsch et al. 2016, are more specifically designed to disentangle the different components of these non-diffusion-dependent noise sources.

4) Clarity of presentation.The Results section is difficult read, and several key aspects of the approach and findings are only clarified during careful reading of the Methods section. Most prominently, there is insufficient explanation of key model predictions that may be counterintuitive for many readers; a lack of clarity around what individual model parameters capture; and confusing elements to how the model fits are presented. We encourage the authors to carefully revise the Results section with this concern in mind.Specific recommendations:a. Implications of AtD vs DtA strategy choice:The fact that, all else being equal, the DtA strategy generates ‘more’ precise behaviour on Computed trials than the AtD model (at least for the parameter range human participants seem to occupy here) is the central feature that differentiates behaviour produced by the two strategies and renders the models identifiable. The authors also seem to take the direction of this effect to be self-evident, as no effort is made to explain to the reader why this pattern emerges. For instance, readers may wonder whether allowing for N = [2, 5] sources of gaussian noise compared to only 1 source should actually produce ‘more’ variability in behaviour. Now, the averaging over particles that takes place at the culmination of the DtA process counteracts the greater total noise to produce less variability in behavioural reports. But this was far from self-evident, and this key effect should be unpacked.

We apologize for the lack of clarity. We have revised the manuscript substantially to address this issue. We now include a new subsection near the beginning of results entitled “Diffusing-Particle Framework and Predictions” that provides intuitions, descriptions, and justifications of our modeling choices. We have also added a table that includes a description of the model parameters and relationships (Table 1).

b. Model parameter descriptions:There seems to be a general lack of clarity around what exactly each model parameter, and in particular different subscripts to different parameters, are supposed to capture. In Figure 2, for example, subscripts N, MN, 1, N(E/L) and MNSeq are all used but only explained in Methods.

We have expanded and clarified our descriptions of the model parameters in the Results section, including descriptions of all of the subscripts. We have also added Table 1 for easy reference.

c. Alternative strategy interpretation. (see also point 3):Please clarify what exactly the finding is, because this currently seems ambiguous: Compare line 244 "(…) participants had roughly equal tendencies to use either of the two strategies" implying that we can distinguish which strategy individual subjects are following, vs. line 257 "(…) neither of which was more likely than the other for a given participant" which implies the opposite.

We apologize for the confusion. We now make it clear that: 1) we can distinguish the two strategies (except when *A*=1), and 2) under some conditions the two strategies had roughly equal prevalence across our participants. The ambiguous sentence has been removed.

d. Model fitting procedures (see also point 2):The role of the A parameter in the different fits is confusing, specifically, seeing fits of A for the AtD model since, this parameter does not seem to used at all in the AtD process. If our understanding is correct, the A parameter in these fits is only relevant to producing behavior in Perceived blocks with set size > 1 – a condition of the experiment to which the AtD process is never actually applied. But at no point is this made explicit, leaving room for quite considerable confusion when the reader encounters this important section of the Results.

Please see our response to point 2 and Specific Recommendations for point 4a. We have made major revisions to the text to make sure the procedure for determining *A* is clear.

[Editors' note: further revisions were suggested prior to acceptance, as described below.]

Essential revisions:1) Please address one outstanding clarification question by Reviewer #1, with which Reviewer #2 agreed. Rather than summarizing, we paste the original reviewer point below. Once this point is addressed, the paper can be accepted without additional review.I only have one lingering point of confusion that I would welcome clarification on. This again centres around treatment of the A parameter in the AtD model. The authors write in the current manuscript (p.7) that in this model "the average is calculated immediately upon observing the evidence and then stored as a single particle in working memory" (lines 95-96) and then "the single estimate held in working memory diffuses with the same diffusion constant as a single perceived item (σMN^2^ = σ1^2^) (lines 97-99). Based on this my understanding is that there is only ever one particle diffusing in the AtD model during Computed blocks, regardless of set size; this particle always has the same diffusion constant (σ1^2^), and there is, therefore, ‘no role’ for set size/the A parameter in determining diffusion noise during Computed-block memory maintenance in the AtD model. Why, then, is it later written that "Because of the previously described relationships between σ1^2^, σN^2^, and σMN^2^ it is therefore also true that in the AtD model σN^2^ = σMN^2^ * N^A^" (lines 109-110)? Given the earlier sentences, the only way I can see this being true is if N here refers to the number of ‘particles’ being maintained in memory (which, in the AtD model, is always equal to 1, and so the N^A^ term is doing no work here and just causes considerable confusion) – and not the set size presented to the participant, as N is consistently used to denote elsewhere in the paper. I'm sorry if I'm missing something here, but this seems a key conceptual point to get right for clear presentation and differentiation between the two models. The new Table 1, and my careful reading of the Methods, seems consistent with my own intuition that A plays no role on Computed blocks in the AtD model. But this seems fundamentally inconsistent with the equation emphasized on lines 109-110; and indeed with the authors' response to point 2 in the first round of reviews, which I must confess I did not understand.Now, assuming my own interpretation is correct, and that indeed the A parameter is not doing any work on Computed blocks in the AtD model (instead, in this model it only serves to set the diffusion noise across different set sizes in ‘Perceived’ blocks), then I stand by my initial point that without clarification, it is misleading to include and invite comparison of fitted A parameters for the AtD and DtA models in the same table (new Table 2). In one case (AtD) the A parameter only captures (and in turn, will only be constrained by) behaviour in Perceived blocks; in the other (DtA) it captures (and is constrained by) behaviour in both Perceived and Computed blocks. But currently, this is never made explicit.

The reviewers wish us to clarify the role of the *A* parameter, particularly with regards to the AtD strategy in Computed blocks. We regret the confusion and very much appreciate the careful reading and the opportunity to clarify this important point.

It is true that *A* is not directly relevant to the execution of the AtD strategy in the Computed condition, because only one value (the average) is remembered. However, *A* is relevant to determining the best-fitting AtD diffusion constant for a Computed item (σ_MN_^2^), because in AtD, A governs the relationship between the diffusion constant of N Perceived items held in memory (σ_N_^2^) and one Computed item held in memory (σ_MN_^2^): σ_N_^2^ = σ_MN_^2^ * N^A^. Our fitting procedure therefore determined the best-fitting value of *A* per participant and set size by finding the value that best enforced this relationship between data from Perceived and Computed blocks. These points are now clarified in lines 109–116.

To further clarify the reviewer’s specific comments: (1) in the above relationship, *N* continues to refer to the set size and number of particles being maintained; and (2) because *A* was determined from both Perceived and Computed data for both AtD and DtA fits, we believe the comparisons we present are useful.